# Uplink vs. Downlink: Machine Learning-Based Quality Prediction for HTTP Adaptive Video Streaming

**DOI:** 10.3390/s21124172

**Published:** 2021-06-17

**Authors:** Frank Loh, Fabian Poignée, Florian Wamser, Ferdinand Leidinger, Tobias Hoßfeld

**Affiliations:** Institute of Computer Science, University of Würzburg, 97074 Würzburg, Germany; fabian.poignee@informatik.uni-wuerzburg.de (F.P.); florian.wamser@informatik.uni-wuerzburg.de (F.W.); ferdinand.leidinger@informatik.uni-wuerzburg.de (F.L.); tobias.hossfeld@informatik.uni-wuerzburg.de (T.H.)

**Keywords:** HTTP adaptive video streaming, quality of experience prediction, machine learning

## Abstract

Streaming video is responsible for the bulk of Internet traffic these days. For this reason, Internet providers and network operators try to make predictions and assessments about the streaming quality for an end user. Current monitoring solutions are based on a variety of different machine learning approaches. The challenge for providers and operators nowadays is that existing approaches require large amounts of data. In this work, the most relevant quality of experience metrics, i.e., the initial playback delay, the video streaming quality, video quality changes, and video rebuffering events, are examined using a voluminous data set of more than 13,000 YouTube video streaming runs that were collected with the native YouTube mobile app. Three Machine Learning models are developed and compared to estimate playback behavior based on uplink request information. The main focus has been on developing a lightweight approach using as few features and as little data as possible, while maintaining state-of-the-art performance.

## 1. Introduction

The ongoing trend in social life to often use a virtual environment is accelerated by the COVID-19 pandemic. Throughout the last year in particular, work, social, and leisure behaviors have changed rapidly towards the digital world. This development finds resonance in the May 2020 Sandvine report, which revealed that global Internet traffic was dominated by video, gaming, and social usage in particular, with these accounting for more than 80% of the total traffic [1], with YouTube hosting over 15% of these volumes.

For video streaming, the Quality of Experience (QoE) is the most significant metric for capturing the perceived quality for the end user. The initial playback delay, streaming quality, quality changes, and video rebuffering events are the most important influencing factors [2,3,4]. Due to the increasing demand, streaming platforms like YouTube and Netflix have had to throttle the streaming quality in Europe in order to enable adequate quality for everybody on the Internet [5]. This affects the overall streaming QoE for all end users, and ultimately the streaming provider’s revenue from long-term user churn.

From the perspective of an Internet Service Provider (ISP), responsible for network monitoring, the goal is to satisfy their customers and operate economically. Intelligent and predictive service and network management is becoming more important to guarantee good streaming quality and meet user demands. However, since most of the data traffic is encrypted these days, in-depth monitoring with deep packet inspection is no longer possible for an ISP to determine crucial streaming related quality parameters. It is therefore necessary to predict quality by other flow monitoring and prediction techniques. Furthermore, the increasing load and different volumes of flows, and consequently the processing power required to monitor each flow, make detailed prediction even more complex, especially for a centralized monitoring entity.

In this scenario, given the enormous number of different streaming sessions and the associated humongous global data exchange, the analysis of every video stream at packet level cannot be carried out. To enable monitoring in a decentralized way with limited resources available in the last mile, a lightweight approach without unmanageable overhead is important to improve current services. This leads to a scalable in-network QoE monitoring at decentralized entities through a proper QoE prediction model.

This work reviews several different Machine Learning (ML) approaches to estimate the initial playback delay, video playback quality, video quality changes, and video rebuffering events using data originating from the native YouTube client on a mobile device. The goal of this work is to investigate the following research questions:
Is estimation of the most important QoE parameters possible using only low-volume uplink data?What data is required and how much prediction accuracy impairment is acceptable with only uplink data, compared to state-of-the-art downlink-based ML prediction approaches?

From our investigation, we show that the Random Forest (RF)-based approach in particular shows promising results for all metrics. Using only uplink information, this approach is lightweight compared to downlink-based predictions. Based on a dataset that was measured over 28 months between November 2017 and April 2020, a total of over 13,000 video runs and 65 days of playback are evaluated. Studying the performance of the models provides valuable insights, helping to draw conclusions on the applicability of specific models and parameter options. The contribution of this work is threefold:Three different general ML models are defined, compared, and evaluated on the entire dataset with YouTube streaming data from the mobile app in order to predict the most common QoE-relevant metrics. It is shown that even a simple RF-based approach with selected features shows F1 scores of close to 0.90 to help decide whether video rebuffering occurs during the download of a video chunk request.Details of ML improvements are discussed. This defines a lightweight approach that only uses uplink traffic information, but with only a marginal decrease in prediction accuracy for a target metric. The learning is based on the fact that during a streaming session, the uplink requests are sent in series to the content server to fill the client’s buffer. After playback starts and the buffer is full, the behavior changes to an on–off traffic pattern, which is characteristic for streaming today. This is utilized in the learning algorithm, since the complete playback behavior is reflected in a change in the inter-arrival times of the uplink requests.The third contribution is to verify that the approach is practically feasible and requires lesser monitoring resources by estimating the processing and monitoring effort for random artificially-generated video content and defining a queuing model for studying the load of a monitoring system.

Each model is validated using videos other than those utilized for testing, which leads to only slightly worse results. As a result, our work is a valuable input for network management to improve streaming quality for the end user. The remainder of this work is structured as follows: in Section 2, background information is defined with a focus on video streaming and streaming behavior used as information in the methodology in this work to predict streaming quality. The streaming quality metrics are also introduced here. Section 3 summarizes related work, first with a literature overview and afterwards through a differentiation between uplink and complete packet trace monitoring, as done in most related works by an effort analysis. Section 4 summarizes the measurement process involving the testbed and the studied scenarios, and Section 5 presents the dataset. A general overview of relevant information in the dataset is provided, with focus on the relevant streaming quality metrics. The dataset is used to train the models presented in Section 6, where additional information about the features and feature sets is given. The results are discussed in Section 7 for each QoE relevant streaming metric. Section 8 concludes the paper.

## 2. Background

This section contains important definitions and essential background information on streaming, especially streaming-related terminology. The basic streaming process is described, as are the individual phases that a streaming process undergoes. Different factors influencing the overall streaming quality are introduced briefly, at the end of this section.

### 2.1. Terminology

Terms and concepts used in this work are defined in this subsection and differentiated from each other. Accordingly, a short description of streaming is followed by explanation of important terms from the network and application perspectives.

#### 2.1.1. Video Streaming

Video streaming is the process of simultaneously playing back and downloading audio and video content. Streaming is defined by the efficient downloading of the content that is currently being played back by the user, which offers two advantages. From the user’s point of view, playback can begin as soon as the content part matching the current play time is loaded. From the provider’s perspective, only the part that is consumed has to be downloaded, and therefore no resources are wasted. For streaming in general, the available end-to-end throughput between content server and user must match the video encoding rate. Any variation in the video encoding rate and network throughput is balanced by a buffer. Playback continues as long as enough video and audio data are available at the end-user’s player, otherwise the stream stops. These video interruptions are called stalling. Thus, video streaming is a continuous request and download process. Appendix A has more details about this process, and streaming in general.

#### 2.1.2. Traffic Patterns

Traffic patterns in video streaming are the mapping of the application video behavior to network traffic. A pattern is a sequence of network packets corresponding to the requested network chunks. Different sequences of network chunks are requested in different playback situations, depending on what the buffer state is and how much data is required at the application level. From this, the application state can be estimated. The understanding of the traffic patterns forms the basis for detecting the current buffer level, and thus the overall player state leading to the playout quality. The time of content requests is calculated and compared based on the traffic patterns. If the inter-request time of video chunks is higher than the amount of video requested with each chunk, the buffer depletes. If the inter-request time is shorter, the buffer is filled up. Consequently, it is possible to draw conclusions to the buffer change, from the uplink request traffic pattern. Different streaming phases are defined in the following paragraphs. Based on the traffic patterns in these phases, the streaming behavior estimation process is done in this work.

### 2.2. Streaming Phase Definition

Streaming phases are states of the internal streaming process. At the user’s end, only the effects and consequences of these phases on the streaming state are noticeable. To the user that manifests as smooth streaming, a quality change, or a video rebuffering. From an application point of view, two different phases in video streaming can be defined according to [6], an initial buffering phase and a periodic buffer refill phase. In the initial buffering phase, video data is requested and delivered in a best effort manner until the target buffer threshold is reached. Then, the player switches to the periodic buffer refill phase. This phase corresponds to the normal streaming behavior, where as much data is downloaded as is extracted from the buffer for playback. The goal here is to keep the buffer slightly above the target buffer threshold. Thus, the next data is only requested and downloaded when the buffer level falls below the target buffer. As a result, there is no video data downloaded as long as there is sufficient buffer. In contrast, the authors of [7] define three phases. The initial buffering is called buffer filling, the periodic buffer refill steady-state. Additionally, a buffer depletion phase is defined where the buffer level decreases, and thus the amount of pre-buffered seconds also decreases.

For a detailed stalling analysis, the chunk request traffic patterns are analyzed in this work for each of the streaming phases defined in [7], together with the pattern when the playback is interrupted and the video stalls. This is essential for an insight as to the phases with higher and lower influence on quality.

### 2.3. Quality Influencing Factors

Since this work deals with video playback quality estimation, this section introduces factors that influence playback quality for the end user. According to [3], the most important quality degradation factors for video streaming, influencing the QoE for the end user, are the initial playback delay, the played out quality and playback quality changes, and stalling.

#### 2.3.1. Initial Delay

Initial delay describes the waiting time between video request and video playback start. According to [8], the initial delay can be divided into two parts, the initial page load time and the time delay between the initial video request and the playback start. While the initial page load time is not only influenced by the currently available network conditions and the video content delivery network (CDN) state and quality, but also, among others, by the smartphone type, quality, and memory usage, this work investigates the delay from initial video request to playback start as initial delay.

#### 2.3.2. Streaming Quality and Quality Changes

Next, the currently played out quality, especially the amount and frequency of quality changes, influences the QoE for a streaming end user. Since, in HTTP Adaptive Streaming (HAS), the goal is to adapt the playback bitrate to the currently available bandwidth, especially in high variability bandwidth scenarios, frequent quality changes are possible. However, since according to [3], a constant though lower quality is preferable over frequent quality changes, but with a slightly higher average quality, the goal is to decrease the degree of quality change to a minimum, while on the other hand increasing the overall average quality. For that reason, in most streaming players and adaptation algorithms, different metrics such as a quality change threshold are implemented, which define specific rules to trigger a quality change.

#### 2.3.3. Stalling

Stalling, the interruption of video playback, is the last metric influencing the video QoE, which is discussed in this section. Since stalling has the highest negative influence on the overall QoE [3], a detailed QoE detection or estimation is based on the possibility of detecting or estimating stalling events. Furthermore, since, according to the literature, the QoE degradation is worse with many small stalling events compared to fewer but longer events [9], it is also important to detect small stalling instances, and short playback phases between stalling events.

## 3. Related Work

There are different methodologies in the literature to quantify the video playback quality, detect and estimate the metrics influencing the Quality of Service (QoS), and at the end predict the perceived QoE for the end user. For that reason, this section covers the literature discussing methods to detect or estimate quality degradation metrics, QoS or QoE measurement or prediction approaches, and different ideas to quantify playback quality. Furthermore, request and feature extraction approaches relevant for streaming based works are presented. An overview of selected, especially relevant, related work is summarized in Table 1. The main influencing factors as reflected in the table, which differentiate approaches, are six: real time applicability, the target platform, the approach itself, the prediction goal, the data focus, and the target granularity from time window based approach to a complete session.

There are many works in the literature describing, predicting, or analyzing the QoE. A comprehensive survey of QoE in HAS is given by Seufert in [3]. Besides knowing the influencing factors, another important goal is to measure all relevant parameters for good streaming behavior. One of the first works tackling streaming measurements calculating the application QoS and estimating the QoE based on a Mean Opinion Score (MOS) scale was published in 2011 by Mok [19]. However, with the widespread adoption of encryption in the Internet, and also in video streaming, the approaches became more complex, and simple Deep Packet Inspection (DPI) to receive the current buffer level was no longer possible.

Nowadays, several challenges must be resolved for in-depth quality estimation in video streaming: First, data must be measured on a large scale. This is already being done for several platforms, including, among others, YouTube [20], Netflix and Hulu [21], and the Amazon Echo Show [22]. Afterwards, a flow separation and detection is essential to receive only the correct video flow for further prediction. Here, several approaches exist for streaming, e.g., [7,23], as ISP middle-box [24], or with different ML techniques [25].

After determination of the correct video flow, the approach is defined. In modern streaming, two approach types are well adapted: session reconstruction and ML models. The session reconstruction-based approaches show that it is possible to predict all relevant QoE metrics. Whereas in [26], only stalling is predicted with session reconstruction, Mangla shows its utility in [17] for modeling other QoE metrics and comparing them to ML-based approaches. Similar works have been published by Schatz [27] and Dimopoulos [28] to estimate video stalls by packet trace inspection of HTTP logs.

Especially in recent years, several ML based approaches have been published for video QoE estimation. For YouTube in particular, according to Table 1, an overall QoE prediction with real time approaches and focus on inspecting all network packets, is available from Mazhar [10], Wassermann [11], and Orsolic [12]. For all works, an in-depth analysis of voluminous network data is done, and more than 100 features each are extracted. While Mazhar only provides information about prediction results for the YouTube desktop version, the mobile version is also considered by Wassermann. Furthermore, Wassermann shows that a prediction of up to 1 s granularity is possible with a time-window based approach, with accuracies of more than 95%. The received recall for stalling as the most important QoE metric is 65% for their bagging approach and 55% for the RF approach. The precision is 87% and 88%, respectively. In this work, we receive macro average recall values of more than 80% and a comparable precision. Furthermore, Orsolic provides a framework that can also be utilized for other streaming platforms.

Different approaches are adopted by, among others, Bronzino [13] and Dimopoulos [14]. Although the two works are not real time capable, Dimopoulos studies the QoE of YouTube with up to 70 features, while Bronzino also trains the presented model with Amazon and Twitch streaming data by calculating information like throughput, packet count, or byte count.

Gutterman [18] has a different approach, in which, compared to the packet based approaches, a video chunk-based approach is chosen, with four states: increasing buffer, decreasing buffer, stalling, and steady buffer. The approach is similar to this work, while the data resolution on application layer is higher in the approach from Gutterman. It is evident that in Gutterman’s dataset the phases with dropping buffer are even more underrepresented, from 5.9% to nearly 7%. In contrast, in the present work it is 9.9%. Furthermore, Gutterman uses sliding windows up to a size of 20, including historic information up to 200 s. Overall, similar results are achieved. When we compare the video phase prediction to the best results of Gutterman’s study that is received for the Requet App-LTE scenario, slightly better precision and recall scores for the stalling phase are achieved by Gutterman. The results for the steady phase and the recall for the buffer increase is similar. In the buffer decay phase (depletion in this work), we outperform Gutterman by more than 6% in the precision score and 50% in the recall score. Nevertheless, with the difference in dataset and the amount of included historic information, a complete comparison is difficult.

While different ML algorithms are investigated, most of the related approaches mentioned use at least one type of tree-based ML model. This popularity is a result of multiple factors, including ease of development and lower consumption of computational resources. Moreover, the tree-based algorithms, e.g., RF, perform on a similar level or better than others [11,13,18], when comparing multiple algorithms within one work.

Gutterman [18] investigates the performance of a NN, with similar results compared to a tree-based approach. Shen [15] uses a Convolutional Neural Network (CNN) to infer initial delay, resolution, and stalling. The real-time approach uses video data from YouTube and Bilibili and predicts QoE-metrics for windows of 10 s length. Among the tested ML algorithms, the CNN achieves the best results. The potential of deep learning to predict QoE is emphasized further in the work of Lopez [16] by combining CNN and Recurrent Neural Network (RNN).

### 3.1. Preliminary Considerations on Monitoring and Processing Effort

The approaches in the literature use different time windows and parameters of uplink and downlink in different granularity. Given the varying degrees of complexity, there is a trade-off between accuracy and computational effort.

This is especially important in terms of predicting streaming video parameters in communication networks on a large, distributed scale. An efficient approach is usually preferred here, which can be adopted on the last mile close to customers where little computing power is available. Hence, the question arises, with regard to the related work above: How much monitoring and computational effort is required to achieve high accuracy with machine learning techniques for video streaming?

In preparation for the following study, we examine how much data might be generated during video streaming of arbitrary videos and how much data needs to be monitored and processed with regard to the various methods available in the literature. While there are a large number of proposed approaches that detect quality, stalling, and the playback phases, the approach in this work focuses exclusively on partial information for monitoring and processing, e.g., the uplink information and the features generated from it.

In the following steps, 9518 random artificially generated videos are created according to models from the literature [29,30], and the effort is quantified that is required to obtain the specified partial information while streaming them (streaming simulation). Note that this is meant to be a pre-study with focus on arbitrary video content. For the dataset used for the actual study later, we refer to Section 4.2 and Section 5 for more information on the dataset and measurement scenarios. The generated data, scripts, and additional evaluation are available in [31]. The aim is to show how much data and effort is required when machine learning is used with partial data in comparison to a full monitoring. Later on, in the next sections, machine learning is carried out that quantifies the accuracy for different feature sets on partial data.

#### 3.1.1. Methodology

The basis for this evaluation is the modeling of different videos with data-rate models generating video streaming content that is then streamed with the help of a simulation of adaptive bitrate adaptation logics (ABRs), which are then mapped to requests at the network layer in order to determine the amount of data while streaming. All traffic models for H264/MPEG-like encoded variable bitrate video in the literature can be broadly categorized into (1) data-rate models (DRMs) and (2) frame-size models (FSMs) [29]. FSMs focus on generating single frames for video content in order to gain the ability to create arbitrary yet realistic videos [30,32]. The approach works as follows: First, we create the video content in different resolutions. This results in a variable bitrate per second video content for each resolution. Afterwards, adaptation is applied to the video content with ABRs. In this case, an adaption of three quality switches is applied with a duration of one third of the total video length. Finally, this can be used to calculate how many bytes must be processed and monitored if (1) the entire video has to be considered or (2) if only 5, 3, or 1 HTTP requests in up- and downlink have to be considered.

#### 3.1.2. Results

Table 2 shows the generated video content according to [30,32]. We have generated a total of 9518 videos from 9518 samples each for every common resolution from 144p to 1080p. Figure 1a shows the empirical cumulative density function (CDF) for the bytes required for streaming a randomly generated video. The individual lines in the plot reflect the effort required to consider the downlink traffic, the uplink traffic, and the entire corresponding traffic. Overall, the 50% quantile results in a traffic reduction of around 86% if only the uplink has to be considered. This shows the advantage of approaches that only require uplink data for monitoring and prediction. As suggested above, these approaches are particularly interesting in practice when there are no large computational resources available; for example, when making predictions close to the user within the last mile.

Figure 1b shows the data effort if only a time window of the last *x* HTTP requests (1, 3, or 5 requests) is used. This is of particular interest for the prediction itself, since only a small amount of data has to be processed here.

### 3.2. Queuing Model of QoE Monitoring Entity

This statement is further evaluated by a simple queuing model studying the monitoring load of a monitoring entity for QoE prediction depending only on uplink traffic, compared to a full packet trace consisting of uplink and downlink traffic. A visualization of the model is given in Figure 2.

#### 3.2.1. General Modeling Approach

During traffic flow monitoring, all packets of a video stream arrive at the monitoring instance, wait until the monitoring entity is free, and are subsequently processed. This is described by a Markov model with a packet arrival process *A*, an unlimited queue, and a processing unit *B*. The arrival process can be described as a Poisson process with rate λ when a sufficient amount of flows are monitored in parallel [33]. This holds true for this approach, since studying a small number of flows does not require this effort. The processing unit *B* processes the packets in the system with rate μ. To analyze the model, a general independent processing time distribution with expected average processing time $B=\frac{1}{\mu }$ can be assumed. This holds true, for example, for a deterministic processing time distribution. Then, the system can be described as an M/GI/1−∞ system. For a stable system λ≤μ is given, which means the arrival rate must not exceed the processing rate, in order to not overload the system. Furthermore, for real-time monitoring or to satisfy specific network management service level agreements (SLA), a sojourn time or system response time t<x can be defined, while *x* describes the delay in the system. For reasons of simplicity and since the model is only used as an illustration, the following model is analyzed as an M/M/1−∞ system with an exponential processing time distribution.

#### 3.2.2. Modeling Results

For the usability study of the approach in a real environment, two different questions are answered: (1) What is the average sojourn time or average response time of the system E[T] dependent on the processing time of the processing unit within the monitoring entity; and (2) How many video streams, or streaming flows, can be monitored in parallel without exceeding a target delay with a probability *q*? Both questions are answered, based on the arrival rates for uplink traffic ($\lambda_{ulsim}=0.32/$s) and the complete packet trace containing uplink and downlink traffic ($\lambda_{allsim}=108.22/$s) received by the streaming simulation above. Furthermore, the uplink traffic rate ($\lambda_{ul}=0.22/$s) and the complete uplink and downlink traffic rate ($\lambda_{all}=79.65/$s) of the complete dataset used for this work are used. A detailed overview of the dataset follows, in Section 5.

To answer the first question, Little [34] is used to receive the average response time by E[X]=λ·E[T], with $E\left[X\right]=\frac{\rho }{1-\rho }$ as long-term average number of customers in a stationary system and $\rho =\frac{\lambda }{\mu }$ as utilization of the processing unit. With Little [34], E[T] is received as $E\left[T\right]=\frac{E[B]}{1-\rho }$ with $E\left[B\right]=\frac{1}{\mu }$ as average processing time of the processing unit within the monitoring entity. As processing time, a value range between 1 s and 1 s is studied in Figure 3a at the x-axis. The y-axis shows the resulting response time of the system. The yellow line shows the result when monitoring uplink and downlink data considering the streaming simulation, whereas the orange line is the result considering the full packet trace received by the dataset. The results are shown in brown when monitoring only uplink traffic for the streaming simulation, and in black for the dataset. It is evident that the average response time is similar up to a processing time of 10 ms per packet. For slower systems, the processing entity of the monitoring system is not able to analyze all packets of the complete stream anymore. Thus, the response time increases drastically. When only the uplink data is to be monitored, the processing entity can process all data up to a processing time of more than 1 s per packet. Thus, slower and cheaper hardware could be used.

To answer the second question of how many streaming flows can be supported in parallel, E[T]<x or P(T<x)=q with x as target processing time or target delay in seconds is introduced. Furthermore, *q* is the target probability describing the percentage of packets that must be analyzed faster than *x* to guarantee specific SLAs. For positive *t*-values, the distribution function for the response time is described by $F\left(t\right)=1-e^{-(\mu -\lambda )t}$. Solving this, the supported positive arrival rate λ is dependent on the target probability *q* according to $\lambda \leq \frac{\ln(1-q)+\mu t}{t}$. Figure 3b shows the amount of flows supported in parallel by a 1 G switch for monitoring only uplink traffic, compared to all uplink and downlink packets based on a target delay *x* in milliseconds. The assumption is that the switch can process 1 Gbps and the packets always fill the maximal transmission unit (MTU) in the Internet, of 1500 B. The target probability *q* is set to 90% and 99.9%, respectively. It is obvious that focusing on only uplink-based monitoring allows 100 to 1000 times more flows compared to full packet trace monitoring. Furthermore, since there is an active trend towards increasing data rates with higher resolutions and qualities, uplink-based monitoring is a valuable methodology to deal with current and future video streams while saving hardware costs or distributing monitoring entities.

## 4. Measurement Process

In-depth QoE monitoring of YouTube adaptive streaming is difficult, especially for a mobile device. For that reason, this section summarizes the measurement approach by introducing the testbed and the measured scenarios.

### 4.1. Testbed

The testbed used for data collection in this work is installed at the University of Würzburg. It consists of a Linux control PC running an Appium server, several Android smartphones running an Appium client, and a Wi-Fi dongle. On the smartphones, videos are streamed via the native YouTube application for Android. Next to the challenge of encrypted network traffic used for YouTube streaming [35], the major challenge is to access the internal client data of the mobile app. Compared to custom made apps, like the YoMoApp [36], the approach presented in this work directly measures the native YouTube app behavior with the Wrapper App [37]. This is done by accessing and controlling an Android smartphone via USB and the Android Debugging Bridge (adb). In order to manipulate the effective bandwidth for a measurement run, the Linux tools tc and NetEm are used to shape the trace on the Internet link of the PC which acts as a proxy for the mobile devices. The bandwidth for the smartphones is controlled accordingly. Note that previous example measurements show a maximum incoming bandwidth of 400 Mbit/s and always sufficient memory and CPU. For that reason, the control PC is not the bottleneck in the streaming process. There is a dummy video between each run, to ensure a full reset of the bandwidth conditions. The quality setting in the YouTube App is set to automatic mode to investigate the adaptive video streaming mechanisms. There are two important sources of data, the network and transport layer traces, which are captured with the tcpdump tool and contain all the network data for a measurement run with a smartphone, and the application layer data, which are obtained by repeatedly polling YouTube’s Stats for nerds. More information about all available application parameters and the detailed Wrapper App behavior are available in [37].

### 4.2. Measurement Scenarios

More than 75 different bandwidth scenarios are applied to study the streaming behavior of the native YouTube app. First, 38 constant bandwidth scenarios are measured to analyze the general streaming and requesting behavior with the app. There, the bandwidth ranges from 200 kbit/s to 2 Mbit/s in 100 kbit/s steps and from 2 Mbit/s to 10 Mbit/s in 500 kbit/s steps. Additionally, constant bandwidth settings of 20 Mbit/s, 25 Mbit/s, and 100 Mbit/s are studied to get insights into best case streaming situations. Second, specific quality change and stalling scenarios are defined, where the bandwidth is lowered during video playback. To force the player to request different qualities, several smaller and larger variations in the bandwidth setting are applied, while scenarios with very abrupt bandwidth degradation to 200 kbit/s or lower are used to trigger stalling in case of a buffer underrun. Third, real bandwidth traces from mobile video streaming scenarios according to [38,39] are used. The measurements show that the 4G traces of [38] are most likely sufficient to stream the video in consistently high quality. Furthermore, no stalling events could be detected applying these scenarios. In contrast, [39] show lower bandwidth scenarios that most likely trigger quality changes. In most measurements, the quality is automatically chosen by the YouTube App so as to not influence the common behavior of the data requesting process. In all, 243 YouTube videos are measured for evaluation. In addition to the 18 videos listed in [26], 225 additional random YouTube videos are measured to cover a broad range of typical content. All data are aggregated to a large dataset that serves as the basis for the machine learning process. The dataset is presented in detail in the following paragraphs.

## 5. Dataset

For an in-depth analysis of the streaming behavior, a dataset is required that covers specific and regular behavior for all QoE relevant metrics, which are initial delay, streaming quality, for YouTube the video resolution, quality changes, and stalling. For that reason, a dataset is created in this work using the above-mentioned testbed and measurement scenarios, which covers 13,759 valid video runs and more than 65 days of total playback time. Parts of the dataset are already published for the research community [8,20]. In the following subsection, first the dataset postprocessing is explained, giving details on network and application data. Afterwards, a dataset overview is presented, with the main focus being on streaming relevant metrics.

### 5.1. Postprocessing

The postprocessing of the application data and network traces in this work is twofold. First, the network traces are analyzed and all video related flows are extracted. Afterwards, the application data are postprocessed and combined information from the application and the network is logged for further usage.

#### 5.1.1. Network Data

The network data are captured in the pcap tcpdump format. By following the IP and port addresses of the googlevideo.com DNS, all traffic associated with the YouTube video can be separated from cross traffic like advertising, additional video recommendations, or pictures. For each packet of this traffic, the packet arrival time, the packet source and destination IP address and port, and the payload size in bytes are extracted.

Consecutive download of video data is required in video streaming to guarantee smooth playback. Therefore, requests are detected in the uplink, asking for video data from the YouTube server. Thus, all packets containing payload to the YouTube server are extracted from the received packets and named as uplink chunk requests. Here, we assume that the video is downloaded in a consecutive manner according to Figure 4. This means request *x* is always requested before request *x* + 1 and downloaded completely. Thus, all downlink traffic following a request in the same flow is marked as associated to that specific request, before the next one is sent to the server. In this way, information such as the amount of uplink and downlink traffic in bytes, amount of uplink and downlink packets, and protocol associated with one request can be extracted. Furthermore, the last downlink packet of request *x*, the end of *downlink duration x* in Figure 4, need not necessarily match with the next uplink request timestamp.

#### 5.1.2. Application Data

The application information Stats for nerds is available as a JSON-based file. There, several application specific parameters are logged together with a timestamp in 1 s granularity. Table 3 gives an overview of all parameters relevant for this work. The postprocessed application data are added to the network data for further usage. The full dataset with all information for each request used in the learning process in this work is available in [31]. Further details about data extraction are presented in [8,20].

### 5.2. Data Overview

After data postprocessing, a more detailed overview of the dataset is visible, as shown in Table 4. The table shows that 7015 of all videos, or 51.0%, have at least one change of the *fmt*-value in Stats for nerds. This change is a playback resolution change according to the YouTube HAS algorithm, also used as quality change. Thus, in the following paragraphs, the terms resolution change and quality change are used interchangeably. The dataset shows 22,015 quality changes in all. This is detectable by having different video format ID values (fmt-values) in a single video run. Furthermore, 2961 videos or 21.5% have at least one stalling event, accounting for a total of 5934 stalling instances and more than 32 h of stalling time. The stalling is detected by analyzing the buffer health and the played out frames. If the timestamp of the log is increasing faster than the frames are played out and the buffer health is close to zero, the player is assumed to stall.

The stalling duration varies from less then one second to 117 s for all videos in the dataset. Short stalling of less than 5 s occurs in 20% of all stalling cases. The medium stalling duration is 10.5 s and the mean is detected at 19.87 s. So as to have no conflict with the initial playback delay, stalling within the first 5 s of video playback time is not considered. Furthermore, for a learning approach with historic information as investigated in this work, stalling prediction at the beginning of the video is not possible.

The dataset contains videos streamed with TCP and UDP, visible in the network data. Among all runs, 22.9% are streamed with TCP, and the remainder with UDP. A more detailed investigation of the main QoE degradation factors initial delay, playback quality, quality changes, and stalling in the dataset follows in the respective sections of the methodology.

To acquire more information about the parameters influencing specific streaming relevant metrics, first of all a dataset study is done. Since no video specific information like video ID or resolution is available from encrypted network traffic, this is not studied in this work. Furthermore, the available downlink bandwidth to stream the videos is not analyzed, since it is not directly detectable from the traffic trace. Note that there is no explicit analysis of the amount of totally downloaded and uploaded bytes for specific QoE relevant metrics, since this is not the main focus of this work and is highly dependent on the video resolution. The focus in the following paragraphs is on the most relevant influencing parameters with an impact on the prediction result. Additional data set analysis is presented in Appendix B.

### 5.3. Initial Delay

By using the first non-zero played out frames value, information is received when the first playback is logged in the Stats for nerds data. Since the logging granularity there is about 1 s, the exact play start cannot be extracted from the data. Using the number of played out frames and the video fps, showing how many frames are played out within one second, the initial playback start can be calculated. Furthermore, the initial video chunk request timestamp is logged in the network data file. That is subtracted from the playback start timestamp to receive the ground truth initial delay. The analysis shows that the mean ground truth initial delay is 2.64 s.

#### Request Study

Since the main objective of this work is estimating QoE relevant metrics based on uplink request, the required number of uplink requests to start the video playback are studied for initial delay prediction. Therefore, all detected chunk requests before the playback starts are summed up. Figure 5a shows in black the number of requests sent to the YouTube server before the playback starts for all videos as CDF. The brown line shows the number of requests completely downloaded according to the network data analysis presented above. The figure shows that the playback can start when only a single request is sent, and for a very little percentage of all videos before the first request is downloaded completely. Furthermore, for more than 60% of all measured runs, five requests must be sent to start the video and in total up to ten requests can be required until the initial video playback starts. Since according to Figure 5a, the number of chunk request downloads started before playback start can be higher than the number of chunk request downloads ended, video playback can start in the middle of an open chunk request download. This happened in 69.72% of all runs. However, no additional information is received by further analyzing this behavior.

### 5.4. Video Quality

For an in-depth analysis of the streaming behavior, next to the initial delay, the video quality is an important metric. The video quality is described by the overall playback quality and the extent of quality changes. Since this work deals with the analysis of playback quality with network data without studying the effect of packet loss, no frame errors like blur or fragments in single frames are observed. Thus, only the YouTube format ID as quality is taken into consideration in this work.

In total, more than 1.1 million requests are measured, as summarized in Table 5. The table shows that, except for the 1444p resolution, more than 80,000 requests are measured each. The highest representative is 720p, which is the target resolution for many videos at the smartphone without triggering higher quality manually. Since only 841 requests are measured for 1440p video resolution and YouTube did not automatically trigger 1440p quality, in the following sections only quality up to 1080p is studied. Furthermore, except for newest generation smartphones, which incidentally also start using 4K videos, most smartphones do not support quality higher than 1080p. A screen resolution of 1440p or higher was used at end 2019 in less than 15% of all phones [40]. For that reason, 1440p video resolutions are omitted from further evaluation.

To gain greater insight into the requesting behavior of the different quality representations within the dataset, the inter-request time of specific resolutions is studied in the paragraphs that follow.

#### Inter-Request Time Study

Inter-request time is the main metric to describe the streaming process in this work. An overview of the inter-request time for all available video resolutions is given in Figure 5b as CDF. The x-axis shows the inter-request time in seconds, whereas the colors of the lines represent the different resolutions, starting at 144p and ending with 1080p, with lighter colors for higher resolutions.

The figure shows that the inter-request time for 10% to 20% of all requests, regardless of resolution, is very small. The measurement shows that this behavior is visible most likely at the beginning of the video, where many requests are sent. For lower resolution, in the best effort phase at the beginning of the video, the data for a single request can be downloaded very fast when enough bandwidth is available, since it is rather small. For that reason, the next request is sent with only a short delay, leading to a short inter-request time. Another explanation for this behavior is the parallel requesting of audio and video data. A more detailed explanation of this is given in [26], although this plays no further part in this work.

Figure 5b shows, for all resolutions, that more than 70% of all inter-request times are between 0.1 s and 12 s. The rather linear distribution for resolution between 240p to 720p in that time interval in particular is visible, with 240p having the lowest gradient. This means that the inter-request times for all quality levels between 240p to 720p, discussed first in the following paragraphs, show a similar distribution, with 240p having slightly fewer smaller inter-request times compared to the others. For 240p, nearly 60% of all requests show an inter-request time of less than 6 s, while for the others it is 65% to 70%.

The 144p and 1080p resolutions show a different behavior. While 144p shows more instances of larger inter-request times, with only 38% smaller than 6 s, more instances of smaller times are visible for 1080p, with 88% being below 6 s. This observation shows the tendency of having larger inter-request times for poorer quality. One reason for this behavior is the different amounts of data that must be downloaded for specific resolutions, with less data for smaller ones. To minimize the overhead, more seconds of video playback can be requested with a single request for lower resolution. Another reason for this behavior is that lower resolution is used during phases with bandwidth issues. There, the next request can be delayed because the download of the previous one is not finished. Additionally, in the course of this work, the downlink duration and the downlink size per request are studied. The results are similar to the inter-request time study, and thus presented in the Appendix C.

### 5.5. Quality Change

The other video quality based metric influencing the received streaming QoE of an end user is quality change. According to the literature, the duration and the volume of quality change are of interest [3]. Furthermore, the difference in quality before and after the quality change is important; that is, whether the next higher or lower resolution is chosen after a quality change or if available resolutions are skipped [41]. A more detailed overview of quality change detected in the dataset is provided in the following lines.

In total, 22,123 instances of quality change are detected. By mapping these quality changes to requests, 1.94% of all requests are marked as requests with changing quality. A quality change is mapped to request *x*, if it occurs between the request of *x* and x+1 according to Figure 4. An in-depth analysis of the quality before and after change is added in Appendix D for interested readers.

### 5.6. Stalling

An overview of the stalling behavior is provided next, as the most important metric influencing streaming quality degradation. Out of the 13,759 valid video runs in the dataset, stalling is detected in 3328 runs, totaling 6450 stalls. In about 50% of the stalling videos, one stalling event is detected, and a second event is detected in another 25% of the runs. Only in less than 3% of all stalling videos are more than 5 stalling events measured. Furthermore, in 40% of all videos where stalling is detected, two or more different qualities were played out when stalling occurred. This means that, although the quality is adapted to the currently available bandwidth, a stalling event occurred. With further regard to the network parameters investigated in cases of stalling, the data show that stalling occurred while starting the download of from one up to 30 requests. This means that stalling continues although other video chunk requests are sent. For that reason, it might be not sufficient to only monitor high inter-request times or delayed requests for stalling detection. More details about stalling position, duration, and distribution across the qualities in the dataset are available in Appendix E.

## 6. Methodology

This section presents the general methodology of this work. For each streaming quality relevant metric discussed in this work, the prediction approach is presented. Initially, an overview of all used features, general ML approach used for all target metrics, and general data postprocessing is given. Afterwards, the initial delay and the quality prediction are discussed in detail to begin with, since the estimation is done without additional information of the streaming behavior. Then, the approach for quality change and stalling prediction is introduced by presenting the idea with video phase detection, prediction, and the correlation to specific uplink request patterns. Prediction specific information for these approaches is given at the end of this section.

### 6.1. Feature Overview

To receive satisfactory and correct results in the ML process, a thorough feature selection is required. The goal of this work is to keep the volume of network layer data and thus processing time required for the analysis as low as possible without losing prediction accuracy. Thus, only uplink traffic and aggregated downlink traffic are used, instead of analyzing each downlink packet. Based on this, an overview of all features used for prediction, with an explanation, is given in Table 6.

The request start timestamp *f1* describes the moment in time when a request is sent to the server relative to the initial request. Thus, the initial request is always set to zero. The *request size* feature *f4* describes the request size of the single uplink packet requesting a new chunk. Note that a description of the other non-trivial features is given in Figure 4. For correct ML behavior, all categorical features are *one-hot-encoded*, especially to avoid unwanted behavior with TCP and QUIC. The ML models for the important streaming quality metrics are presented based on the features, in the paragraphs that follow.

### 6.2. Feature Set Creation

The prediction quality is highly influenced by, among other things, a thorough feature selection process. For that reason, several feature sets are created for each prediction goal, with three main objectives: The full feature set for prediction with all available features, the *selected* feature set using the features with the highest importance scores after an initial importance ranking, and the *uplink* feature set with only uplink information. An overview of all feature sets for all predicted metrics and the associated features is given in Table 7.

#### 6.2.1. Full Feature Set

The full feature set contains all relevant features for each prediction case, and thus all required for the initial delay prediction. To predict the other metrics, the request start and the port are excluded. For the tasks, the duration is more important compared to the actual timestamps of the request starts. While the port information could be important for detecting server changes, the feature importance for the port feature is investigated, but achieves a low score. Therefore, the port information is discarded, as the port is usually randomly selected and therefore the ML model may overfit with this feature.

#### 6.2.2. Selected Feature Set

To create the *selected* feature set, SelectKBest [42] is used to receive a feature importance score with the *full* feature set as input. The result of the SelectKBest algorithm for all predicted metrics is presented in Table 8. The *mutual_info_regression* score function is used for the regression based initial delay prediction, whereas for all classification based predictions, the *mutual_info_classif* score function is used.

The table shows that no clear best features are visible for the initial delay and the quality prediction. Thus, for the initial delay, all features of the full feature set except the port and the protocol are used. A test run is done for the quality prediction, without the inter-request time (f2) and the downlink duration (f3), because of the low scores. Since the result shows 10% lower accuracy values compared to the full feature set, and the features are essential for the uplink feature set, both features are included again.

For the streaming phase, the quality change, and stalling prediction, the best features are clearly distinguishable from the other features. Accordingly, the inter-request time (f2), the downlink duration (f3), and the downlink (f5) are used.

#### 6.2.3. Uplink Feature Set

To answer the research question, if the most important QoE influencing metrics are predictable with only uplink based information, for all metrics an uplink feature set is created. The least number of features are required for the initial delay, according to Table 8. In the course of this work also additional feature sets were tested to show the potential of this approach. Unfortunately, the same prediction accuracy could not be achieved for the other QoE-related metrics with this limited number of features. Especially for the quality prediction, it was not possible to achieve satisfactory results. The result is an accuracy of less than 65% with only uplink based features. Thus, the downlink duration (f3) is added to the uplink features in the uplink feature set as a trade-off between prediction accuracy and data usage. Only the timestamp of the last downlink packet received for each request is required, to know the downlink duration. For that reason, only the timestamp of a single downlink packet for each request must be kept for prediction. For the other metrics, all uplink features of the full feature set are aggregated to the uplink set.

### 6.3. General Data Postprocessing

For accurate prediction, the data must be prepared as input for the ML in several steps.

#### 6.3.1. Bootstrapping

Class balancing is used to balance the dataset in such a way that it contains equal data points for each target output value. For example, the stalling feature is 0 for most data points due to the fact that the videos are in a playback state most of the time and stalling events are rare in comparison. ML models achieve better results if the training dataset is balanced. Data points are randomly sampled from the target values using bootstrapping, with progressively fewer occurrences, until eventually the dataset is balanced.

#### 6.3.2. Shuffling

Another step is shuffling. This helps prevent problems arising from dependency on consecutive data points in the original dataset. Some ML tasks update their models before one iteration over the training data is completed. Therefore, the initial updates in particular suffer if the data is not shuffled. As a result, similar, statistically dependent data are fed into the model. Moreover, due to the possibly large datasets, even algorithms that in theory process the whole training set at once may be implemented to split and process mini-batches on a machine, due to resource limitations. Thus, shuffling ensures that each data point creates an independent change on the model.

#### 6.3.3. Normalization

Normalization of features and targets is done as a last step. While tree classifiers like RF do not suffer from unnormalized data, NNs do. Nonetheless, the numeric stability that is achieved through normalization benefits certain evaluation and analysis metrics. Therefore, normalization is also used for the RF task. Scikit-learn’s MinMaxScaler determines the minimum and maximum values for each feature and scales them accordingly between 0 and 1.

#### 6.3.4. Test and Trainings Set

In order to reproduce results for the scientific community, it is a good practice to use fixed random seeds for a test- and trainings-set split. However, the split becomes deterministic with a fixed random state. Therefore, during evaluation, different seeds are tested to ensure that the results are valid for different splits. Furthermore, two dataset splits are performed: First, the random trainings- and test-dataset split is 80:20. Second, the test set only contains videos that are not included in the training set, to avoid fitting the model to video specific features. This means that the algorithm trains the behavior on different videos than those it is tested on.

These two datasets are not sufficient if hyperparameter tuning is used to find the best parameter configuration for the models. In the case of a RF prediction, three-fold cross-validation is used in this work. This means that each configuration is trained on two-thirds of the test dataset and evaluated on the remaining third. This is repeated for each split, so each configuration is evaluated three times in total for a combined result. The best configuration is then evaluated with the test dataset for a final result.

For the NN, the dataset is split into 60% training data, 20% validation data, and 20% test data. The validation set is necessary for hyperparameter optimization, because the hyperparamerer tuner Keras Tuner [43] does not support cross-validation for Keras models at the time this work is being written. Without an additional validation set to evaluate each configuration during hyperparameter tuning, the test set would be fitted through hyperparameter tuning, which is unintended. The goal is to improve a general model, not the score on the test set. Therefore, an evaluation of a ML model must always use data which is new to the model. For the LSTM model, an 80:20 split for training and test data is done.

#### 6.3.5. Hyperparameter Optimization

In order to find the best parameter configuration in a given parameter space, hyperparameter optimization is used for the RF approach. There, with the built-in hyperparameter optimization in scikit-learn, a grid search [44] is performed with the F1 score as the target metric. The result of the optimization, and also the hyperparameters used for the prediction, are summarized in Table 9. All other hyperparameters are set to the default values.

For the NN, hyperparameter optimization with Keras Tuner is used. An exhaustive search for model parameters is not possible due to limitations in computational resources and time. The hyperband tuner is used since it employs early stopping, which can save time by discarding bad configurations early without adding another layer of complexity. The set objective is accuracy, the seed is 42, and the maximum number of epochs to train for a single configuration is 50. Each trial configuration is executed twice. The batch size is 1024. After determining the best configuration by accuracy, the model is trained for an additional 50 epochs, resulting in a model that has trained for 100 epochs.

No hyperparameter optimization is done for the LSTM approach because of high complexity and resource demands.

### 6.4. Machine Learning Approaches

Three different ML approaches to prediction are studied in this work: RF, NN, and LSTM. The implementation of the approaches is available in [31]. The following paragraphs give general information on the approaches, especially for the RF approach, since it is used similarly for all metrics. Afterwards, details of the prediction of each specific QoE relevant metric are presented. An overview of the used approaches for each metric is shown in Table 10.

#### 6.4.1. Random Forest

According to the literature [45], good results are achieved with streaming and QoE related challenges when training with RF based approaches. The *sklearn* library implemented in Python is used for creation in this work. Videos are first split into training and test set, the model is instantiated with different random tree generation seeds, and the input features are selected. After selection of the hyperparameters according to Table 9, the model is fitted and evaluated.

#### 6.4.2. Neural Networks

The model for the NN is built with TensorFlow [46] and Keras. To build a NN model in Keras, the layers of the network must be defined, including the number of neurons and the activation function. NNs were studied in detail for all metrics, but showed worse results compared to the RF approach. Furthermore, the training duration is longer and more resource-intense. To show the general approach, the initial delay prediction with NNs is presented and the prediction idea for the streaming phase, the quality change, and the stalling prediction is highlighted in detail hereafter.

#### 6.4.3. Long Short Term Memory

The LSTM-based prediction is applied for the video phase, quality changes, and stalling. Especially for stalling prediction, rather good results are achieved. However, keeping in mind the high complexity and long training times, and the goal of a lightweight, fast, and easy approach, the overall performance is worse compared to the RF approach. Another drawback of both NN and LSTM approaches is the blackbox model.

#### 6.4.4. Initial Delay Estimation

It is assumed that only a specific time period at the beginning of the video download is relevant for good accuracy and performance for initial delay prediction. All requests that are not followed by a data download are removed before prediction commences, since they achieve no change in the video buffer status. The result after this step is a list of all requests with associated feature information of the complete video. This is especially important for the initial delay prediction, since in this case only a limited number of requests are used at the beginning of the video. No prediction is possible if no information is available. Only the initial 20 requests are used for prediction. Subsequently, the influence of a different number is tested.

#### 6.4.5. Neural Network

The input of the NN is a 20 × 11 matrix with 20 requests and 10 features for each video. For the protocol feature, one matrix row is created for TCP and one for QUIC. The workflow of the deep learning model is shown in Figure 6.

The NN consists of a tensor reshaping, adding one dimension to feed two CNN layers to extract high level features. The first layer uses 32 filters with a size of 8 × 2 and a stride of one. The second one has 64 filters instead of 32. The output of each CNN is normalized by a batch normalization layer and reshaped, merging the first two dimensions together. The result is fed into an LSTM layer to learn long-term dependencies between time steps and sequence data. The LSTM layer consists of 100 units, thus providing an output of length 100. To prevent overfitting, a dropout layer is added where a fraction of 0.2 input units are set to 0. Finally, a fully connected layer with 128 nodes and another dropout layer with a fraction of 0.4 follow. The output layer of the network is another fully connected layer. Furthermore, every layer except the last one uses rectified linear units (ReLUs) as activation functions. The RMSprop algorithm is used for optimization, and the Huber loss is used as the loss function.

#### 6.4.6. Quality Estimation

As a next step in overall streaming behavior estimation, the video quality is estimated. Therefore, all requests of the dataset are used as input.

#### 6.4.7. Preprocessing Steps

All requests with quality change are removed for the quality estimation, since no single quality can be predicted there. Furthermore, so as to avoid errors with quality change and stalling events, no request with a maximal buffer filling level smaller than 10 ms is considered. The dataset contains a total of more than 1.14 million usable requests. Before prediction, a standard scaling is done according to [47] to avoid unexpected or unwanted behavior. Afterwards, the RF-based prediction approach is used only as described above. The other models showed less accurate results and therefore are not included in this work.

#### 6.4.8. Phase Detection

The general approach, to achieve a deeper understanding of the streaming behavior, is to analyze the streaming sessions with video phases according to [26]. Based on different traffic patterns, three streaming phases are introduced: buffer filling phase, steady-state phase, and buffer depleting phase. To describe the complete video behavior, an additional stalling phase is introduced for longer time periods with stalling. The traffic behavior within each phase is determined and discussed, and the potential for stalling and quality change is described, in the following subsections.

#### 6.4.9. Buffer Filling Phase

The buffer level in seconds is increasing, during the buffer filling phase. This is common in the initial filling phase of the video, after a quality change, or in case of bandwidth fluctuations after an increase of available bandwidth. Video segments are downloaded in the buffer filling phase in a best effort manner. This means that the complete available bandwidth is used to download video chunks until the buffer is filled. Thus, the average inter-request time of video chunk requests is smaller than the duration of each chunk, until the buffer is filled. The time for one download is directly coupled with the currently available download bandwidth. Thus, faster download is achieved with higher available bandwidth. Since in this phase the potential downlink is higher then the required downlink, no stalling can occur. Furthermore, no change to a lower quality is meaningful. Nevertheless, changes to higher quality are possible. As soon as the buffer is filled completely, the player switches to the steady-state phase, which keeps the buffer at a constantly high level.

#### 6.4.10. Steady-State Phase

Streaming in general is regular downloading and playout of data. If the buffer is filled to the target buffer level, regular chunk requests are required to keep the buffer at a constant level. This behavior takes place in the steady state phase. Theoretically, the bandwidth available in this phase is higher or equal to the required video rate. Thus, a regular pattern of uplink requests can be detected in the uplink direction. In this phase, the buffer is filled enough. Consequently, no stalling event is possible. The possibility of a quality change is also lower compared to the buffer filling phase, since it is expected that a change to higher quality would be triggered earlier, whereas a change to lower quality is not necessary.

#### 6.4.11. Buffer Depletion Phase

In case of bandwidth degradation, the inter-request time between two chunks increases. If the mean inter-request time of a chunk request download is longer than the video playback time the chunk involves, the player is in the buffer depleting phase. In this phase, the available buffer stored in the player decreases during playback. If the duration of this phase is too long and the buffer drops below the quality change threshold, the player requests a lower quality if possible. If the player is already requesting the lowest quality, no change to lower quality is possible. Then, if the buffer runs empty, the playback interrupts and stalling occurs. This also happens if the bandwidth degradation is too abrupt and the buffer depletes to zero, although a quality change is possible and is triggered. Consequently, the buffer depletion phase is of high interest for accurate stalling estimation. Since this is the only phase where the buffer size decreases, a change to lower quality or stalling can occur.

#### 6.4.12. Stalling Phase

The last phase discussed in this work is the stalling phase. This occurs when the buffer has dropped to zero and the video stalls. No significant amount of data is downloaded, here. To exit this phase to the buffer filling phase, it is possible that new data are requested, and therefore single requests are expected. As soon as the bandwidth is increased again or a quality change occurs to adapt the currently available bandwidth to the required bitrate, the player exits that phase and starts to buffer again. A statistical overview of the phases with regard to the requesting behavior is summarized in Appendix F for the interested reader.

#### 6.4.13. Sliding Window Creation

Sliding windows are created, to predict the video phase, quality changes, and stalling. Instead of predicting an output based on a single request, the input is extended to contain the last n requests. Using a sliding window of size n increases the input features by the factor of n. The models can therefore learn on previous requests and how important the historic information is. Furthermore, models are trained with different sliding window sizes to determine how much historic information is mandatory for accurate prediction.

It is essential to balance the dataset prior to feature selection and scoring, as otherwise the chosen features contain a bias towards the larger classes. In the following steps, univariate feature selection with SelectKBest based on mutual information scoring is used on a large sliding window size to determine a smaller but suitable window size for the phase detection task. After creating a sliding window of size ten, the SelectKBest algorithm is performed. Size ten is used as an upper bound because an online model would need to wait for ten requests before a prediction is possible. In the dataset, the average time until the tenth request is 16.1 s. Such a delay is considered high for an online application. Fortunately, the request rate is higher than average at the beginning of a video; therefore, reducing the sliding window size results in a disproportionate decrease in delay, e.g., for a sliding window of five, the resulting average delay is only 5.9 s, which is more acceptable. Accordingly, a sliding window size of five is used in this work for a good trade-off between the delay for prediction start and the accuracy. Smaller values show less accurate prediction results, with a reduction in the F1 score of 1.5% for a window size of 4, more than 2% for a window size of 3, and more than 6% for a window size of 2 and the full feature set.

### 6.5. Prediction

Three ML approaches are used in this work for predicting the streaming phase, quality changes, and stalling. The specifics of the NN and LSTM based approaches for these metrics are explained in the following paragraphs. The RF based prediction is similar to the above. For quality change and stalling prediction, the streaming phase is added as an additional feature to all feature sets shown in Table 7. The results of prediction with and without the streaming phase information are compared.

#### 6.5.1. Neural Network

For each feature set of the NN, the best model is determined via hyperparameter tuning with hyperband. The search architecture consists of two hidden layers, with ReLU as activation function. The search space for each hidden layer is from 128 neurons to 2048 neurons, with a step of 128, and the minimum dropout value is 0 with steps of 0.1 up to a dropout of 0.3. In between, there is a dropout layer with a dropout value of 0.1, to avoid overfitting by deactivating 10% of the neurons randomly during each step of the training. This prevents the network from adapting to a few relevant paths with high weights. Afterwards, the final layer uses a softmax function to generate a probability distribution for each of the four output neurons. The Adam optimizer is used with the learning rates 0.01, 0.001, and 0.0001. For this work, the loss-function is the categorical crossentropy function for multi-class tasks and sparse categorical crossentropy for binary categorical tasks. In the case of multi-class prediction, the last layer has one neuron for each possible output class, and a softmax activation function is used which transforms the output to a probability distribution. Therefore, the sum of all output values of the output neurons is 1, representing the certainty of the model. The output with the highest probability is chosen as the prediction output. Each trial configuration is executed twice.

#### 6.5.2. Long Short Term Memory

The general approach for the LSTM has only small differences from the NN models. First, the sliding window approach is not necessary because the model is built to process time-series data. Therefore, the input is a sequence with an output value for each time step. For each request the LSTM generates a prediction. The second change in the input shape is padding. All input sequences to an LSTM must have the same length. However, there are different numbers of requests for different videos. To solve this problem, a fixed value is appended to stretch the length of each sequence until they match 504, which is the maximum request length in the dataset. The padding value is −10, as there are no negative values in the dataset. Therefore, it is possible to mask the padded values during training and evaluation. When masking is used, the padded values are ignored during calculation of the updates and evaluation metrics. The fixed length limits the model to sequence lengths of 504, but it is possible to apply the approach to longer videos by adding more padding. Bootstrapping is not possible as a single sequence is used as an input to the network and is not split. A sequence already has a class imbalance in itself. Therefore, class weights are used. Samples of rare classes receive large weights, while samples of an overrepresented class receive small weights. Thus, when calculating the loss for updating the model, the model weighs the loss for the different samples in a fashion, which results in an equal contribution from each class to the model updates.

The LSTM model is as follows: First, there is a masking layer to mask the padded inputs, followed by two fully connected layers with 512 neurons each and ReLU as activation function. Then, an LSTM layer with 512 units is used. The return sequences parameter is set to *True* to get an output for each request in the sequence. This is an advantage over the sliding window approach, which is missing outputs for the first requests until the window size is filled. Finally, the output layer uses a softmax function and four neurons to output a probability distribution for each of the four possible phases. Adam optimizer is used with a learning rate of 0.001. Each model is trained for 100 epochs with a batch size of 128.

## 7. Evaluation

This section presents the results for predicting the most important QoE metrics: initial delay, video quality, quality change, and stalling. For quality change and stalling prediction, the results for video phase prediction as the basis are presented according to Section 6.

### 7.1. Initial Delay

In this section, the performance of the introduced approach is evaluated by first presenting the result of the RF learning approach, followed by the performance and challenges of the deep learning model.

#### 7.1.1. Random Forest Model

To study the performance of the RF model, three different initial settings can be varied: the videos in the training and test sets, the tree generation in the RF algorithm, and the features themselves. For that reason, four scenarios are defined according to Table 11. For all scenarios, a random video split in training set and test set is done according to the above-mentioned approach. Using a fixed seed of a specific value, the run produces repeatable, reproducible results. By using random seeds for the video split, the videos used for training and testing are varied, and thus the behavior and possibly the model performance change. Furthermore, since the problem of learning an optimal decision tree is known to be NP-complete [48], different tree generations can be tested to determine the performance variation. This is done by varying the sample of the features to consider, when looking for the best split at each node of the tree. The *video seed* of Table 11 describes the variation in the video distribution in test and training set, and the *tree seed* describes the tree generation as mentioned above.

In the baseline scenario, both video and tree seed are fixed to a specific value (42 in this case). In the three variation scenarios, either the video or the tree seed or both are set to random. For each variation scenario, 15 repetitions are done to see differences in the performance of the initial delay estimation. Thus, in total 45 runs are compared to the baseline scenario. Furthermore, all single runs are evaluated to investigate the performance of any random split compared to the baseline scenario.

From a detailed examination of all 45 scenarios, no significant improvement to the baseline scenario of any specific run can be detected. Moreover, the results in Table 11 show that the mean absolute error (MAE), the 75% percentile, and the 90% percentile are comparable for all variations. It is shown that variation 3 with random test and tree seed performs a little better than the others. However, since the difference is slight, this is not analyzed in detail in the following steps.

It is evident, therefore, that using random seeds in the model creation does not make for significantly better performance in this case, and thus the fixed value of 42 for both the video and the tree seed is used for better and active reproducibility. Note that with other seed values, an additional minor performance improvement may be possible. However, studying this is not the main focus of this work. On the other hand, the improvement of another seed could be a drawback with another dataset.

The next study focuses on the number of requests used for RF learning, where the number of requests varies between 5, 10, 15, and 20. The influence of this variation on the initial delay estimation is presented in Figure 7a as CDF of the prediction error in seconds. The line in black for 20 requests is the baseline of the earlier study. Training the model with 15 requests is shown in brown, 10 requests in orange, and 5 requests in yellow. It is evident that the learning performs similarly for 10 to 20 requests. For 5 requests, a discernible performance loss is presented. The MAE for the scenario with 5 requests is 0.96 s. Compared to that, for 10, 15, and 20 requests it is 0.68 s, 0.65 s, and 0.70 s, respectively. That being the case, the following models are trained with 10 requests as the new baseline scenario, since there is no significant performance loss compared to the other scenarios while using a minimum number of requests.

##### Estimation Performance

After studying the influence of a random test and training set split, in this section a manual split is introduced according to Section 6.4.4. Thus, in the following paragraphs, the baseline scenario with 10 requests is compared to a manual split in test and training videos. The test set in the manual split contains no videos which the model is learned on. Thus, this study shows valuable results for generalizability. The performance is presented in Figure 7b. The *unknown video* scenario, represented by the brown line, shows the manual split with completely different videos used for test and training sets. Thus, the model predicts the initial delay for completely unknown videos. This is compared to the baseline scenario in black. To quantify the quality of the results, the *mean error* line in orange and the *mean error unknown videos* line in yellow are added. The *mean error* describes the estimation error when always estimating the initial delay, with the real mean initial delay value of 2.64 s as the benchmark. The *mean error unknown videos* line shows the same, but for the prediction with only unknown videos in the test set.

The results show that the prediction always performs better than when simply estimating the mean initial delay. The baseline scenario shows a MAE of 0.68 s, while the *mean error* scenario has a MAE of 1.62 s. Furthermore, it is shown that in more than 65% of all cases, even with unknown videos, the model can predict the initial delay with a prediction error of less than 1.0 s. For more than 80%, the error is less than 1.5 s. The MAE for predicting the initial delay with unknown videos is 1.00 s, and the median prediction error is 1.06 s. The differences in the prediction error range of 2.0 s and more are a result of different splits in test and training sets. Nevertheless, similar results are achieved if the videos in the test set are exchanged. This approach outperforms always predicting the overall mean initial delay, in more than 85% of the cases. Hence, the approach is also valuable for videos that were not measured in this work and which do not exist in the dataset. In addition, the estimator performance is studied when using other feature sets for a RF prediction approach. The goal was to minimize the number of features to a minimum. This study is added as Appendix G.

#### 7.1.2. Deep Learning Model

In order to provide a more general statement about the initial delay estimation based on uplink requests with ML methods, the results achieved from the deep learning approach presented in Section 6.4.4 are discussed in the following paragraphs.

##### Model Performance Analysis

For examining the behavior of the model, the prediction performance with an input of 5, 10, 15, and 20 requests is studied. Therefore, the model is trained for 300 epochs, since this value showed good results without overfitting. More than 120 training and testing reruns are done for each study. The MAE achieved for all runs is analyzed. Note that each single result of the study is the MAE of a complete run with 120 training and testing reruns. Thus, single initial delays are estimated with smaller or larger errors.

First, the results show that the number of requests has an influence on performance. For 5 requests, the MAE over all runs is close to 1 s, with a smaller variance and a maximal value of 1.6 s. For 10 requests as input, some predictions show a MAE of about 0.5 s and more than 50% show a MAE of more than 1.5 s. A similar performance is achieved for 15 and 20 requests, while the MAE is smaller for a larger percentage of runs. In more than 60% of all runs, 15 requests as input show a MAE of about 0.5 s; for 20 requests it is nearly 90%. On the other hand, some runs show a MAE of more than 1.5 s. This result is different from that of the RF approach. There, the MAE for several reruns of one specific scenario is always in the range of less than ±0.1 s; as shown, for example, in the initial model setting comparison. The result variance in the deep learning approach is larger, with maximal values in the MAE of single runs of several seconds not applicable.

The prediction performance on unknown videos is studied next, with completely different results, depending on the videos in the test set. For one specific video set, the model performs much better than in the random video case, with most MAEs at about 0.5 s, whereas with other videos in the test set, the MAE is always at about 1.5 s or above. Thus, we assume that this behavior is the reason for large variance in the random video case. Furthermore, we conclude that performance with the presented model is highly dependent on the videos used for testing, and therefore the model is not useful for this prediction without any improvement.

Studying the prediction performance with different, reduced feature sets shows only little improvement, without the uplink bytes and downlink packets. Overall, no statistically significant improvement is visible in the results, either when leaving out single features or when training the model with other feature sets; see Appendix G, Table A2 for more information. The same is obvious when adding an additional convolutional layer, which also leads to much longer training duration. We conclude, therefore, that either the dataset is too small for a comprehensive initial delay investigation based on a NN, or that the NN approach performs worse than the RF approach on the uplink data based prediction. To conclude, though, this does not mean that a RF based approach always outperforms NNs for streaming prediction. An in-depth model tuning and feature importance analysis can influence the estimator performance, although in many cases this requires much longer training duration and is much more complex. This, however, was not the main aim of this work, the objective being rather to create and use a simple and lightweight approach.

### 7.2. Quality Estimation

The results for the video quality estimation according to Section 6.4.6 are presented next with the *full-*, the *selected-*, and the *uplink-*feature sets of Table 7.

#### 7.2.1. Full Feature Set

The correct quality for the full feature set is predicted with 84.3%. The full confusion matrix is presented in Table 12. The highest value is determined for predicting 720p quality. There, prediction accuracy, precision, recall, and F1 score are 92%. The worst performance is for 240p, with a prediction accuracy of 75%, a precision of 75%, a recall of 62%, and an F1 score of 68%. The reason for the large prediction difference is the nature of the different qualities. While the qualities are more distinguishable based on the downlink size for larger qualities compared to 144p, 240p, and 360p, the differentiation, especially between 720p and 1080p, is clearly visible by the inter-request time, as shown in Figure 5b. This is less clear for the resolutions between 240p and 480p.

The different number of requests of each resolution influence slightly the overall prediction quality. The macro average precision is 0.81, compared to 0.84 for the weighted average precision. The macro average recall is 0.72 and the macro average F1 score is 0.74. In comparison, the weighted average precision and F1 score are 0.84. This little imbalance cannot be counterbalanced by bootstrapping, since during hyperparameter optimization, using bootstrapping for the requests did not increase the overall performance. An overall prediction accuracy of 84.68% is achieved with hyperparameter optimization. The macro and weighted average values for precision, recall, and F1 score are similar to the prediction without bootstrapping. Furthermore, it is evident that in most mis-predictions the adjacent resolution is predicted. The error for non-adjacent resolutions is less than 10% for all resolutions.

Furthermore, no statistically significant differences in prediction results are achieved when using only unknown videos in the test set. It is assumed that the video information has no significant impact on the prediction and is thus not relevant.

#### 7.2.2. Selected Feature Set

For the *selected* feature set, a prediction accuracy of 83.11% is achieved. The weighted average precision, recall, and F1 score are at 0.83, while the macro average precision is 0.79 and the macro average recall and F1 score is 0.78. This shows a minor class imbalance due to the higher percentage of 720p-resolution requests. Compared to the estimation with the full feature set, though, the macro average values are improved. Thus, it is assumed that removing less relevant features reduces overfitting without losing overall prediction accuracy.

#### 7.2.3. Uplink Feature Set

For the uplink feature set, results regarding precision, recall, and F1 score for all resolutions are shown in Table 13 to present the influence of the reduced feature set on the prediction quality for each resolution. It is evident that for the 720p resolution, the prediction still works rather well, whereas for the 240p resolution and the 360p resolution, the prediction is much worse compared to the full feature set. This is so especially for the 240p quality, where in 24.23% instances the 144p resolution is predicted. A similar result is seen for the macro average precision with 0.75, recall with 0.73, and F1 score with 0.74. The weighted average values are 0.79 each. Thus, it becomes clear that the uplink feature set is not enough to predict each resolution accurately. For low resolutions in particular, this does not work, although for high resolutions, especially 720p in this case, the results are good. For the 720p quality, the accuracy, recall, and F1 score are at 0.89 and the precision is 0.88.

### 7.3. Video Phase Estimation

The video phase prediction according to Section 6.4.8 is presented next. This is essential to receive details about the status of the video player, if for example enough bandwidth is available, and thus enough data for the player to keep the buffer at a steadily high level. The results for the approaches and all feature sets are summarized in Table 14. The top of the table shows the feature sets and the performance metrics listed in the columns. The column at extreme left shows the prediction, in this case either the phase, the macro average score, or the weighted average score. The ML approach is listed in the second column.

#### Prediction Accuracy

In the RF case, the results show large differences, depending on the predicted phase. The filling and steady phases especially are predicted with good results and F1 scores of 0.94 and 0.90, respectively, for the full feature set. The result is different for the depletion phase and the stalling phase, with F1 scores below 0.80. In the depletion phase, most prediction errors predict the steady phase. For the stalling phase, most mis-predictions are of the depletion phase or the filling phase. The difference in the macro and weighted average scores is a result of the class imbalance in the nature of streamed videos, with their having many more requests in the steady and filling phases compared to the stalling and depletion phases. This could be one reason for the difference in the prediction result. For the LSTM-based prediction, the results are similar for the full feature set.

Furthermore, the table shows that the different feature sets perform similarly in the phase detection case. The selected feature set shows slightly worse results compared to the full feature set. The result for the uplink based feature set improves compared to that of the selected feature set. Thus, in contrast to the video resolution prediction presented above, the uplink-based feature set is convenient for predicting the current video phase, and shows that uplink-based data is enough for phase prediction.

The LSTM-based approach shows slightly better results for the selected feature set compared to the RF approach. Especially for the depletion phase and the stalling phase, slightly better results for recall, F1 score, and macro avg are visible. The same, though less clearly, is evident for the uplink feature set. Thus, the LSTM-based approach can handle the underrepresented classes better. However, for the much more complex algorithm tuning and longer and more resource intense runtime, the difference is very small.

When using only unknown videos in the test set, the results are similar to the other phase prediction experiment for both approaches, with some limitations; again, there is no large difference between using the full feature set, the selected feature set, or the uplink based feature set. Furthermore, the filling phase and steady phase prediction works rather well, with only slightly worse precision, recall, and F1 score values compared to the *known video* experiments. The limitation is in predicting the depletion and stalling phases, where precision, recall, and F1 score values of 0.5 or less are achieved. The goal for future work is to improve this result by increasing the percentage of requests in the underrepresented classes.

The results for the NN approach are worse compared to the other estimations. The best prediction is achieved for the filling phase, with F1 scores of more than 0.9 for all feature sets. The other phases, especially the depletion and stalling phases, are predicted much worse, with an F1 score of less than 0.75 for both phases for the full feature set. The overall macro average F1 score is 0.8 or lower for all feature sets. Especially in the stalling phase, the depletion phase is often predicted falsely.

### 7.4. Quality Change Estimation

Several studies are carried out to predict the quality changes. The best results, as summarized in Table 15, are achieved with a random test- and training-set split and the correct video phase as additional input feature. The table shows that the prediction of no quality change works very well, with F1 scores of 0.97 and higher, both for the RF and the LSTM approaches for all feature sets. Note that the results listed with a score of 1.00 show high 99% results. Since the negligible differences in the high 99% range have no influence on the overall statement, no more digits are provided after the comma.

In contrast, predicting the requests with quality change does not work so well. Different results are seen in those cases for both ML approaches. While for the RF-based prediction, the precision at about 0.80 is rather good, the recall shows values only slightly above 0.50. For the LSTM-based approach, the results are vice versa, with results of only about 0.30 precision. This is explainable by the common streaming behavior, where not many quality changes are visible compared to the total playtime according to Section 5.5. For that reason, the dataset is not balanced, with 22,123 requests with quality change out of a total of more than 1.1 million requests. Second, a downward shift to lower quality is usually triggered in the middle of depletion phases or after stalling. However, there is also change to higher quality during filling phases. This variety of situations where quality changes can be triggered makes prediction difficult.

A more detailed observation of the results reveals that both ML approaches have benefits, based on the prediction goal. The high precision in the RF based approach shows a high true positive rate, and thus most predicted quality changes could be identified as such. The drawback is that many are missed. The high recall for the LSTM-based approach shows that many quality changes are detected, though with the drawback of falsely classifying many requests as quality change requests.

Two additional experiments are done for the quality change prediction: using only unknown videos in the test set and predicting quality change without knowing the video phase during prediction. The results show that knowing the video phase before predicting quality change increases the prediction result by up to 3% for both approaches. As such, an initial video phase determination is essential for good quality change prediction.

Not knowing the video has a significant influence on a metric with previously lower score. This is the recall for the RF and the precision for the LSTM-based approach. For a more general approach with unknown videos in the test set, a more detailed study is required. Moreover, according to the phase prediction, a more balanced dataset could be valuable. This can also help to improve performance in the NN approach, where the results were much worse compared to the others.

### 7.5. Stalling Estimation

Video stalls are estimated in this subsection, as the last but most important QoE metric. The prediction results, with the approaches presented in Section 6, a random test- and training-set split, and without video phase as input feature, are summarized in Table 16. The general table structure is retained as earlier. The *no stalling* line describes the prediction result for all requests that did not stall, and the *stalling* line for all requests that did stall. It is evident here, again, that precision, recall, and F1 score for the *no stalling* case are higher compared to the *stalling* case for both models. This, also, is a result of the higher number of *no stalling* requests compared to *stalling* ones. However, it is also obvious that regarding the *stalling* prediction, both approaches perform similarly for the *full-* and the *selected-*feature sets. For the *uplink-*feature set, the RF-approach performs slightly better. The precision in particular is lower in the LSTM-based prediction. Additionally, regarding the F1 score, it is evident that similar results are achieved for all feature sets with the RF-based approach, whereas the performance of the LSTM-based approach decreases with fewer features in the feature sets. To conclude, it is possible to predict stalling events at network layer with uplink requests only with an F1 score of 0.89 with a RF approach. In addition, for 60% of all false positives in case of the RF-based prediction and for nearly 80% in the LSTM-based approach, a video buffer of less than 9 s is detected, independent of the used feature set.

Other experiments show that the current video phase is also a valuable input to increase the prediction performance. However, since in this work the *stalling phase* is defined as one of the streaming phases, the input of the phases is meaningless. Nevertheless, the exclusion of buffering and steady phases, which are detected quite accurately, offers high potential to improve the overall performance. With only unknown videos in the test set, the macro average F1 score drops to 0.67 or less. The precision and recall values for the stalling case in particular are lower for all experiments with only unknown videos in the test set. Here, again, improvement potential is visible upon including the streaming phase.

The prediction quality of the NN approach is comparable to the above scores. Overall, a macro average F1 score of 0.85 is achieved for the full feature set, while the stalling class only shows an F1 score of 0.71. The other feature sets show worse results, with a macro average F1 score of only 0.81 for the uplink feature set. The class imbalance, again, is assumed to be the reason for the poorer performance in those cases.

## 8. Conclusions

In this work, the most relevant QoE metrics, i.e., initial playback delay, video streaming quality, quality change, and video rebuffering events are studied with a large-scale dataset of more than 13,000 YouTube video streaming runs watched using the native YouTube app. Three ML models are developed and compared to estimate the playback behavior based on uplink request information. The main focus was to develop a lightweight approach using as few features as possible while retaining state-of-the-art levels of performance.

The results show that a simple RF approach outperforms a NN on the given dataset. With the NN, the variance in the resulting estimation error is larger, while for some runs it is comparable to the RF approach. Nevertheless, large outliers and long training time make it useless as a lightweight approach. The presented LSTM-based approach shows similar performance for the investigated metrics, but with higher complexity and computational effort. The presented RF approach, however, performs reasonably well for all important QoE metrics for a known testing set, and only slightly worse for an unknown set. Especially when using only uplink based data and reducing the feature sets, only a marginal decrease in estimation performance is discernible. For example, for stalling, as the most important quality impairment metric for video streaming, a macro average F1 score of 0.84 is achieved. High recall values of nearly 0.9 show that most requests predicted as stalling are predicted correctly. Additionally, for 60% of all false positives, a video buffer of less than 9 s is detected. In comparison to related state-of-the-art works, where in most cases downlink based approaches or full packet traces are used for QoE prediction with stalling detection rates of 90–95%, the approach is valuable, especially from a monitoring effort point of view, with acceptable prediction accuracy impairment. The good result with a simple RF approach is especially possible by an in-depth study of the streaming process and feature selection. Given this valuable information and a good choice of input features and hyperparameter optimization, we have excellent pre-conditions for good learning.

Accordingly, we conclude that it is possible, based on uplink requests only, to estimate the main influencing metrics for the perceived QoE for the end-user of YouTube videos, decreasing the amount of required data drastically from full packet traces with uplink and downlink data to only chunk requests, inter-arrival times in the uplink data, and a few additional features dependent on the predicted metric only. The goal in future work will be to investigate the generalizability of the approach for other streaming platforms and live streaming.

## Figures and Tables

**Figure 1 sensors-21-04172-f001:**
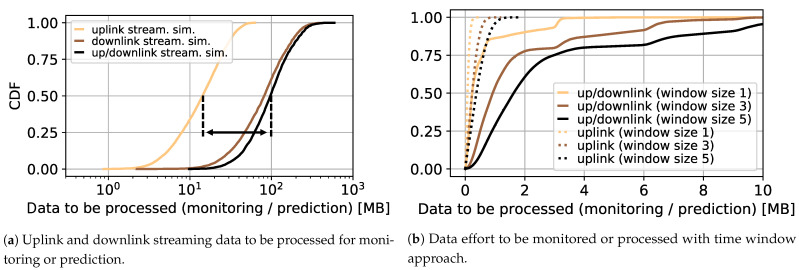
Streaming simulation results.

**Figure 2 sensors-21-04172-f002:**
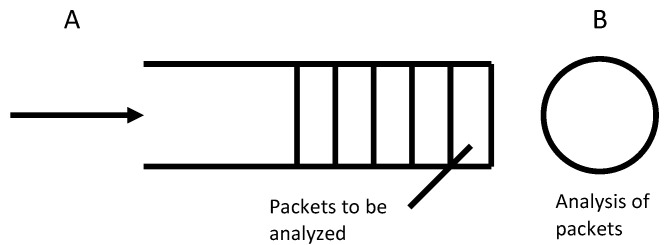
Visualization of queuing model to quantify the monitoring load for the presented prediction approach.

**Figure 3 sensors-21-04172-f003:**
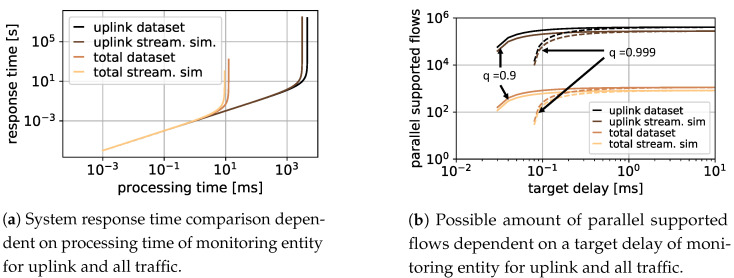
Modeling results overview.

**Figure 4 sensors-21-04172-f004:**
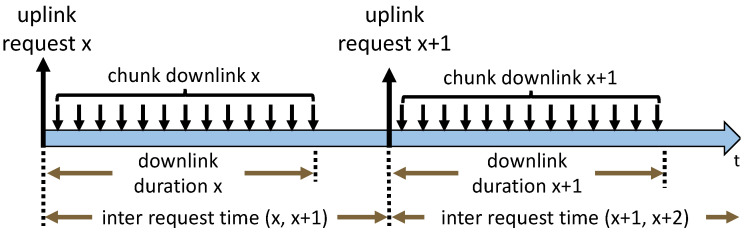
Timeline of YouTube chunk requests and associated data download.

**Figure 5 sensors-21-04172-f005:**
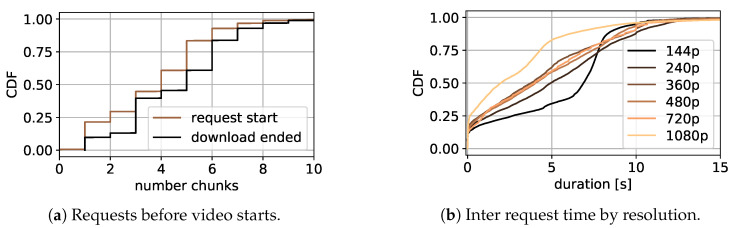
Dataset study: initial delay and video resolution.

**Figure 6 sensors-21-04172-f006:**
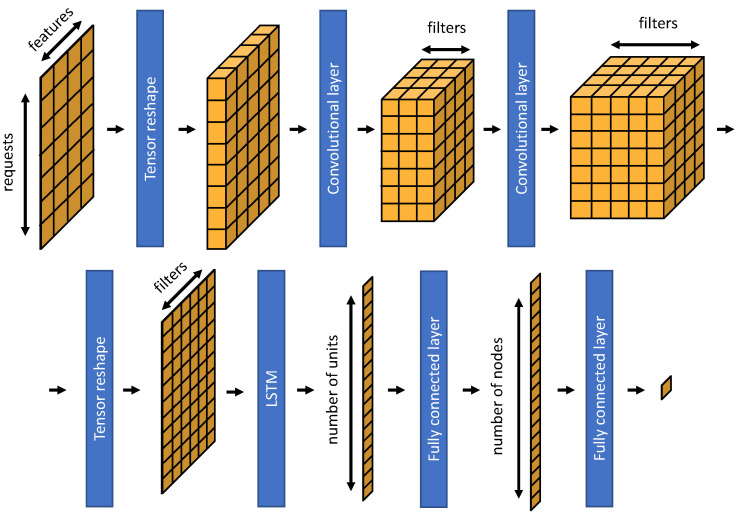
Exemplary visualization of the deep learning model.

**Figure 7 sensors-21-04172-f007:**
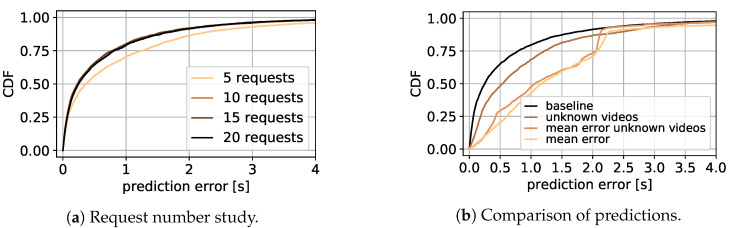
Initial delay prediction results.

**Table 1 sensors-21-04172-t001:** Overview of select related work.

Reference		Real Time	Target Platform	Approach	Prediction Goal	Data Focus	Granularity
Mazhar’18	[10]	✔	YouTube desktop	tree-based	QoE	packet	time-window
Wassermann’20	[11]	✔	YouTube mobile, desktop	RF	QoE	packet	time-window
Orsolic’20	[12]	✔	YouTube desktop	tree-based	QoE/KPI	packet	session
Bronzino’19	[13]	✘	YouTube, Netflix, Twitch desktop	RF	initial delay, resolution	packet	time-window
Dimopoulos’16	[14]	✘	YouTube desktop	RF	stalling, quality, quality changes	packet	session
Shen’20	[15]	✔	YouTube, Bilibili desktop	CNN	initial delay, resolution, stalling	packet	time-window
Lopez’18	[16]	✔	-	CNN, RNN	QoE	packet	time-window
Mangla’18	[17]	✘	-	session modeling	QoE	packet, request	session
Gutterman’20	[18]	✔	YouTube mobile, desktop	RF	buffer warning, video state, video quality	request	time-window
Our work’21	–	✔	YouTube mobile	RF, Neural Network (NN), Long Short Term Memory (LSTM)	QoE	request	time-window

**Table 2 sensors-21-04172-t002:** Feasibility study: artificially generated video content; video duration: 5–15 min.

Resolution/		Bitrate [kbps]		Size [MB]
# Samples		min/⌀/max	Q10/Q90	min/⌀/max
144p	9518	32/114/637	70.2/167.6	2/8/20
244p	9518	175/256/749	212.3/310.1	7/19/33
360p	9518	489/570/1092	526.5/624.2	19/41/69
480p	9518	974/1057/1546	1013.5/1111.1	38/79/124
720p	9518	2091/2172/2661	2128.9/2225.2	79/160/249
1080p	9518	4575/4657/5183	4613.3/4710.2	173/338/529

**Table 3 sensors-21-04172-t003:** Application data overview.

Value	Explanation
timestamp	timestamp of measurement log
fmt	video format ID (itag)
fps	frames per second
bh	buffer health
df	dropped frames and played out frames
videoID	ID of played video

**Table 4 sensors-21-04172-t004:** General dataset overview.

Parameter	Value
all runs	13,759
total runs quality change	7015
amount quality changes	22,015
total runs stalling	2961
total stalling events	5934
average video length	6 min 45 s
total number requests	1,142,635

**Table 5 sensors-21-04172-t005:** Video quality overview.

Resolution	Total Requests	Percentage
144p	156,031	13.93
240p	85,221	7.61
360p	109,223	9.75
480p	184,963	16.51
720p	487,717	43.53
1080p	96,516	8.61
1440p	841	0.08

**Table 6 sensors-21-04172-t006:** Feature overview.

Index	Feature	Explanation
f1	request start	relative request timestamp
f2	inter-request time	time between two requests
f3	downlink duration	request to last downlink packet
f4	request size [byte]	size of request packet
f5	downlink [byte]	cum. downlink for request
f6	downlink packets	packets downloaded for request
f7	uplink [byte]	cumulated uplink for request
f8	uplink packets	uplink packets for request
f9	port	server port of video flow
f10	protocol	used network layer protocol

**Table 7 sensors-21-04172-t007:** Overview of all feature sets.

Metric	Full	Selected	Uplink
initial delay	f1–f10	f1–f8	f1, f2, f4, f7
quality	f2–f8, f10	f2–f8	f2–f4, f7, f8, f10
streaming phase	f2–f8, f10	f2, f3, f5	f2, f4, f7, f8, f10
quality change	f2–f8, f10	f2, f3, f5	f2, f4, f7, f8, f10
stalling	f2–f8, f10	f2, f3, f5	f2, f4, f7, f8, f10

**Table 8 sensors-21-04172-t008:** Feature importance scores (high scores for selected feature sets in bold).

Metric	f1	f2	f3	f4	f5	f6	f7	f8	f9	f10
initial delay	**0.55**	**0.44**	**0.36**	**0.40**	**0.45**	**0.43**	**0.35**	**0.34**	0.05	0.13
quality	–	**0.15**	**0.19**	**0.28**	**0.64**	**0.36**	**0.33**	**0.25**	–	0.11
streaming phase	–	**0.93**	**0.89**	0.21	**0.76**	0.19	0.28	0.12	–	0.15
quality change	–	**0.59**	**0.59**	0.06	**0.49**	0.09	0.17	0.05	–	0.09
stalling	–	**0.60**	**0.60**	0.24	**0.55**	0.16	0.26	0.14	–	0.23

**Table 9 sensors-21-04172-t009:** Hyperparamters for the Random Forest based prediction.

Metric	Bootstrap	Criterion	max_features	min_sample_split	n_estimators
initial delay	True	gini	sqrt	5	2000
quality	False	entropy	auto	5	500
streaming phase	True	gini	auto	2	30
quality change	True	gini	auto	2	30
stalling	True	gini	auto	2	10

**Table 10 sensors-21-04172-t010:** Machine learning approaches.

Metric	RF	NN	LSTM
initial delay	+	+	−
quality	+	−	−
streaming phase	+	+	+
quality change	+	+	+
stalling	+	+	+

**Table 11 sensors-21-04172-t011:** Model variation overview.

Scenario	Video Seed	Tree Seed	MAE	75%	90%
baseline	fixed	fixed	0.68 s	0.79 s	1.82 s
variation 1	fixed	random	0.68 s	0.81 s	1.83 s
variation 2	random	fixed	0.68 s	0.80 s	1.81 s
variation 3	random	random	0.65 s	0.80 s	1.77 s

**Table 12 sensors-21-04172-t012:** Quality prediction result—full feature set.

	144p	240p	360p	480p	720p	1080p
**144p**	**0.7949**	0.1095	0.0231	0.0268	0.0350	0.0108
**240p**	0.0935	**0.7524**	0.0618	0.0465	0.0316	0.0141
**360p**	0.0236	0.0451	**0.7785**	0.0971	0.0433	0.0124
**480p**	0.0148	0.0234	0.0732	**0.7703**	0.0860	0.0323
**720p**	0.0079	0.0066	0.0118	0.0341	**0.9181**	0.0215
**1080p**	0.0100	0.0088	0.0100	0.0365	0.1060	**0.8287**

**Table 13 sensors-21-04172-t013:** Quality prediction—uplink feature set.

	144p	240p	360p	480p	720p	1080p
**144p**	**0.8443**	0.0466	0.0213	0.0319	0.0483	0.0076
**240p**	0.2423	**0.5305**	0.0611	0.0804	0.0737	0.0120
**360p**	0.0507	0.0439	**0.6554**	0.1510	0.0861	0.0128
**480p**	0.0370	0.0248	0.0615	**0.7244**	0.1280	0.0243
**720p**	0.0206	0.0075	0.0127	0.0443	**0.8932**	0.0217
**1080p**	0.0320	0.0124	0.0178	0.0738	0.1220	**0.7420**

**Table 14 sensors-21-04172-t014:** Phase prediction result.

		Full Feature Set			Selected Feature Set			Uplink Feature Set		
		**Precision**	**Recall**	**F1**	**Precision**	**Recall**	**F1**	**Precision**	**Recall**	**F1**
**depletion**	**RF**	0.76	0.81	0.79	0.74	0.77	0.75	0.74	0.78	0.76
	**LSTM**	0.74	0.84	0.79	0.75	0.83	0.79	0.71	0.84	0.78
**filling**	**RF**	0.95	0.93	0.94	0.94	0.92	0.93	0.95	0.93	0.94
	**LSTM**	0.96	0.93	0.95	0.96	0.93	0.95	0.96	0.92	0.94
**stalling**	**RF**	0.77	0.75	0.76	0.72	0.69	0.71	0.77	0.74	0.75
	**LSTM**	0.73	0.85	0.79	0.71	0.84	0.77	0.68	0.85	0.76
**steady**	**RF**	0.89	0.91	0.90	0.86	0.89	0.88	0.87	0.90	0.89
	**LSTM**	0.89	0.91	0.90	0.89	0.91	0.90	0.90	0.90	0.90
**macro avg**	**RF**	0.84	0.85	0.85	0.82	0.82	0.82	0.83	0.84	0.83
	**LSTM**	0.83	0.88	0.85	0.83	0.88	0.85	0.81	0.88	0.84
**weighted avg**	**RF**	0.91	0.91	0.91	0.89	0.89	0.89	0.90	0.90	0.90
	**LSTM**	0.92	0.91	0.91	0.91	0.91	0.91	0.91	0.91	0.91

**Table 15 sensors-21-04172-t015:** Quality change prediction result.

		Full Feature Set			Selected Feature Set			Uplink Feature Set		
		**Precision**	**Recall**	**F1**	**Precision**	**Recall**	**F1**	**Precision**	**Recall**	**F1**
**no quality change**	**RF**	0.99	1.00	0.99	0.99	1.00	0.99	0.99	1.00	0.99
	**LSTM**	1.00	0.95	0.97	1.00	0.95	0.97	1.00	0.95	0.97
**quality change**	**RF**	0.84	0.56	0.67	0.82	0.54	0.65	0.83	0.55	0.66
	**LSTM**	0.30	0.86	0.45	0.31	0.82	0.46	0.31	0.83	0.45
**macro avg**	**RF**	0.91	0.78	0.83	0.90	0.77	0.82	0.91	0.78	0.83
	**LSTM**	0.65	0.90	0.71	0.65	0.89	0.71	0.65	0.89	0.71
**weighted avg**	**RF**	0.99	0.99	0.99	0.99	0.99	0.99	0.99	0.99	0.99
	**LSTM**	0.98	0.95	0.96	0.98	0.95	0.96	0.98	0.95	0.96

**Table 16 sensors-21-04172-t016:** Stalling prediction result.

		Full Feature Set			Selected Feature Set			Uplink Feature Set		
		**Precision**	**Recall**	**F1**	**Precision**	**Recall**	**F1**	**Precision**	**Recall**	**F1**
**no stalling**	**RF**	0.99	1.00	1.00	0.99	1.00	0.99	0.99	1.00	1.00
	**LSTM**	1.00	0.99	0.99	1.00	0.99	0.99	0.99	0.99	0.99
**stalling**	**RF**	0.78	0.70	0.74	0.73	0.66	0.70	0.77	0.71	0.74
	**LSTM**	0.62	0.92	0.75	0.59	0.93	0.72	0.54	0.94	0.69
**macro avg**	**RF**	0.89	0.88	0.87	0.86	0.83	0.85	0.89	0.85	0.87
	**LSTM**	0.76	0.97	0.83	0.79	0.96	0.86	0.77	0.97	0.84
**weighted avg**	**RF**	0.99	0.99	0.99	0.99	0.99	0.99	0.99	0.99	0.99
	**LSTM**	0.99	0.99	0.99	0.99	0.99	0.99	0.99	0.99	0.99

## Data Availability

Parts of the dataset used for this work are published in [8,20]. Details about the measurement process are given in [26,37].

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
