# Peer review of "Uplink vs. Downlink: Machine Learning-Based Quality Prediction for HTTP Adaptive Video Streaming"

_sensors, 2021, doi:10.3390/s21124172_

Round 1

Reviewer 1 Report

This paper examines the most relevant Quality of Experience metrics, i.e. the initial playback delay, the video streaming quality, video quality changes, and video rebuffering events, with an extensive data set of more than 13 000 YouTube video streaming runs that were collected with the native YouTube mobile app. Three Machine Learning models are developed and compared to estimate playback behavior based on uplink request information. The paper is well-formatted but the description of the method is not very clear. It is surprising that the results show that a simple Random Forest approach outperforms a NN on the given dataset. Considering the great achievements of deep learning in many fields such as computer vision and speech recognition, it is hard to believe that the simple random forest methods can achieve such results. Is this a general conclusion, or is it only applicable to uplink and downlink scenarios? It is suggested that the authors carefully analyze the difference between the uplink/downlink scenarios and the general deep learning scenario, and clarify what characteristics make RF achieve such good results. If possible, it is highly encouraged that the author discloses the source code so that the method of the paper can be better understood.

Author Response

Dear Reviewer #1

We would like to thank you for your excellent comments and the time you spent evaluating our paper. We really appreciate your efforts, which have helped to improve the paper. Your comments regarding the neural networks are valid and have been corrected. We have added a brief discussion at the end of Section 7.1.2 and in the conclusion. We have also improved the readability of the text. In particular, the introduction was revised in detail by a professional language editor from the US. Finally, our scripts and datasets are now publicly available to other researchers. In the following, we will go into each point of your review individually and provide answers.

C01: “The paper is well-formatted but the description of the method is not very clear.”

Answer: We fully agree with this comment. The method - and, above all, why it performs so well in comparison to neural networks - was not described in sufficient detail in the text. The amount of information about the streaming process and the dataset has drawn focus away from the methodology. Consequently, we first improved the description of the methodology by shortening the less relevant dataset considerations and background sections, and moved parts to the appendix for further reading. 

In addition, we have added important information and a conclusion in Section 7.1.2 Model Performance Analysis to clarify the usability of random forest-based approaches compared to neural networks (see the highlighted text in the document). We will also try to improve this point again for the camera-ready version. Furthermore, please also have a look at C02 regarding RF vs. NN.

C02: “It is surprising that the results show that a simple Random Forest approach outperforms a NN on the given dataset. Considering the great achievements of deep learning in many fields such as computer vision and speech recognition, it is hard to believe that the simple random forest methods can achieve such results.” 

Answer: Yes, indeed this part was not explained clearly enough in the work. 

In general, by preprocessing of the dataset and an in-depth feature selection, we learned a lot about the streaming process before the machine learning approach was designed. We studied and selected the features in detail by analyzing the streaming behavior and important influencing parameters (such as inter-request time, request size). With this valuable information and an appropriate choice of input features as well as hyperparameter optimization, we have created excellent pre-conditions for a good learning. Thus, at the end, even a simple random forest can predict the target metrics to a sufficient degree. Please note that neural networks could also perform at least as good as random forest based approaches in this case. For neural networks, however, much more configuration and tuning effort would be required with only marginal benefit, see also LSTM evaluations. This is especially visible with the LSTM approach in our work, where we receive comparable results to the random forest. 

It is not the main focus of this work to optimize and finetune learning models; we rather want to highlight that the lightweight uplink-based approach and simple ML algorithms provide sufficiently good quality in certain scenarios. 

We have revised the text to make our contribution clearer and to emphasize that additional fine-tuning can improve the algorithms. 

C03: “Is this a general conclusion, or is it only applicable to uplink and downlink scenarios? It is suggested that the authors carefully analyze the difference between the uplink/downlink scenarios and the general deep learning scenario, and clarify what characteristics make RF achieve such good results.”

Answer: Please see also the answer to the previous comment which partly addresses this comment at the end. 

The objective of this paper was not to provide a general conclusion for machine learning approaches and video streaming. Furthermore, we don't want to suggest that a random forest is always better than a neural network for video streaming. Rather, the focus is on the methodology and the consideration of uplink request data during the streaming phase with an arbitrary machine learning approach. The main benefit from using uplink data is the lightweight monitoring at acceptable performance and good accuracy. Thus, these good results are received with the approach in the paper. We have revised the paper to better emphasize these points. We hope this cleared up the question for you.

C04: “If possible, it is highly encouraged that the author discloses the source code so that the method of the paper can be better understood.”

Answer: Thank you for this suggestion. We fully agree and are happy to upload the data. We uploaded the source code of the learning and the processed datasets for other interested researchers [1]. We have added a readme file with more details inside the dataset. 

We hope that these revisions have adequately addressed your comments.

[1] https://zenodo.org/record/4890859#.YLZa0WYzb0o

Reviewer 2 Report

The authors proposed an evaluation of several ML models to verify their new approach that considers only the uplink traffic in the correlation of QoS and QoE for streaming media.

I suggest opening the dataset (or data traffic generator) for turns possible reproducibility and increases the potential of the paper's citations.

Unfortunately, the initial testbed is fragile to represent the real data behavior. Despite considering the literature to create the data traffic, it is not sufficient to reproduce a real CDN with its distributed processing and autonomous systems (hops and links). On the final data generated by the authors, a real behavior is inspected by the authors, which increases the contributions but does not solve the initial limitations.

I suggest the authors reduce the occupancy of terms and definitions well-known in a short subsection. The size of the paper turns the reading tiresome. Consider shifting unnecessary analysis to and appendices and reducing the number of sections.

Author Response

Dear Reviewer #2

Thank you for your apt comments and the time you spent evaluating our paper. We really appreciate your efforts, which have already helped to improve the paper. We have, and will try again for the camera-ready version, to collect more data, in particular to show that the dataset represents a reliable sample of video streaming. Furthermore, the entire paper was revised in detail by a professional language service from the US. Finally, we moved parts of the background and description to the appendix to improve the readability of the paper. In the following, we will go into each point of your review individually and provide answers.

C01: “I suggest opening the dataset (or data traffic generator) for turns possible reproducibility and increases the potential of the paper's citations.”

Answer: Thank you for this suggestion. We fully agree and are happy to share the data. We uploaded the generated video data with the required scripts, the processed dataset of all YouTube measurements and all machine learning scripts to [1]. Please see the readme files in the uploaded .zip file for more details. 

C02: “Unfortunately, the initial testbed is fragile to represent the real data behavior. Despite considering the literature to create the data traffic, it is not sufficient to reproduce a real CDN with its distributed processing and autonomous systems (hops and links). On the final data generated by the authors, a real behavior is inspected by the authors, which increases the contributions but does not solve the initial limitations.” 

Answer: We agree on this point. In order to reflect streaming with the effects of a real CDN with its distributed processing and its autonomous systems, we used and analyzed measurements from two different locations in Europe (Würzburg, Germany and Paris, France). There, on a high level with respect to the main streaming performance indicators (e.g., initial loading time, quality selection), we saw comparable streaming behavior in both locations. More information is further available in Karagkioules et. al. [23] and Seufert et. al. [39] where details about the measurement process and detailed dataset information is given. We have added this information about [23] and [39] in the paper literature. These papers also discuss the testbed setup in both regions in detail. We will try for the camera-ready version to collect more data, in particular to show that the dataset represents a reliable sample of video streaming.

C03: “I suggest the authors reduce the occupancy of terms and definitions well-known in a short subsection. The size of the paper turns the reading tiresome. Consider shifting unnecessary analysis to and appendices and reducing the number of sections”

Answer: Thank you for this suggestion. Indeed, this improves the readability a lot. We moved information about the streaming process from the background section, the additional dataset overview from the dataset/methodology section, and additional initial delay information to the appendix A-G. In the main paper, we focus now only on the most relevant information regarding our approaches. 

We hope that these revisions have adequately addressed your comments.

[1] https://zenodo.org/record/4890859#.YLZa0WYzb0o

Reviewer 3 Report

The paper starts with an exhaustive study  from a Youtube data set about the influence of streaming features into the prediction/estimation of several key streaming performance metrics. This thorough study allows identifying the “best” features to enter to a machine learning (ML) approach devoted to estimate the metrics. Afterwards, a random forest (RF), a neural network (NN) and a LSTM approaches are designed using the insights obtained in the aforementioned study, and later compared each other. The authors find that the RF, in comparison with the other approaches, is better suited to implement a lightweight ML to be placed at the video client based on local uplink data. This methodology is then adequate, thorough and technically correct.

However, the paper is excessively long (40 pages) in comparison with the final outcome. Most part of the tests do not show conclusive results, which authors actually mention. Very roughly, only the patterns of inter request time and downlink durations seem to be significant to predict/estimate streaming phase, quality changes and stalling, only random forests seem to be efficient to predict/estimate any of the performance metrics. All the tests not showing clear results could be omitted from the manuscript and, eventually, maybe, referred to an online appendix. Moreover, the authors devote the 10 first pages to a tutorial about adaptive video streaming and video quality of experience. A knowledgeable reader (the target public of the paper) should know all this background. A very short and concentrated recall of a few pages and proper quotes to the bibliography should be enough. In conclusion, the paper could be drastically shortened, which actually would improve its readability.

Finally, authors do not perform any comparison with any of the previous machine learning mentioned in the bibliography. Authors aim to demonstrate that a lighter uplink data-based approach is competitive in comparison with already proposed heavier downlink (or full packet traces) based approaches. To test some of the state of the art proposals using the same authors’ Youtube data set will allow to appreciate the contribution of the present work.

Author Response

Dear Reviewer #3

Thank you for your positive and excellent comments and the time you spent evaluating our paper. We really appreciate your efforts in helping to make the paper better. We have improved the readability of the text. In particular, first, (1) the text was revised in detail by a professional language editor from the US. (2) Second, we have changed the structure in the sense that we have moved large parts, as noted by you, to the appendix. This has also been noted by other reviewers. In the following, we will go into each point of your review individually and provide answers.

C01: “However, the paper is excessively long (40 pages) in comparison with the final outcome. Most part of the tests do not show conclusive results, which authors actually mention.”

Answer: Thank you for this suggestion. Indeed, reducing the length of the work improves the readability a lot. We have moved information about the streaming process from the background section, the additional dataset overview from the dataset/methodology section, and additional initial delay information to the appendix A-G. In the main paper, we focus now only on the most relevant information regarding our approaches. 

C02: “Very roughly, only the patterns of inter request time and downlink durations seem to be significant to predict/estimate streaming phase, quality changes and stalling, only random forests seem to be efficient to predict/estimate any of the performance metrics.“

Answer: We agree that only the patterns for inter-request time and downlink durations are significant and other information can be moved to the appendix (see comment C01). You are right that especially the random forest shows good and efficient results. We also added deep learning results to compare the approach to related works and evaluate the performance of different prediction approaches. Especially the LSTM approach shows similar performance compared to the random forest but with much higher effort and fine tuning. Since the comments of another reviewer focused on the deep learning part, if you agree, we decided not move the deep learning results to the appendix but keep the LSTM results as performance comparison to the random forest within the main paper. However, significant parts about the background and the dataset investigation are moved to the appendix. 

C03: “All the tests not showing clear results could be omitted from the manuscript and, eventually, maybe, referred to an online appendix.”

Answer: Agreed. We decided to move all non-relevant dataset evaluations and the additional initial delay investigation to the appendix (see comment C01).

C04: “Moreover, the authors devote the 10 first pages to a tutorial about adaptive video streaming and video quality of experience. A knowledgeable reader (the target public of the paper) should know all this background. A very short and concentrated recall of a few pages and proper quotes to the bibliography should be enough. “

Answer: We reduced the background information to the most relevant ones concerning our approach. This results in (1) a very short introduction to video streaming, (2) in relevant traffic patterns for our approach, and (3) in quality influencing metrics as discussed in the evaluation. The other information is moved to the appendix as reference for less knowledgeable readers.

C05: “In conclusion, the paper could be drastically shortened, which actually would improve its readability.”

Answer: Many thanks. This assumption proved to be correct. The actions we took are listed in the comments above. We reduced the number of pages from 40 to 31 pages.

C06: “Finally, authors do not perform any comparison with any of the previous machine learning mentioned in the bibliography. ”

Answer: Thank you for that comment. We are sorry that the comparison with the reference approaches was not clear enough. We added a detailed comparison of the stalling prediction result as the most relevant QoE metric, to the state of the art packet based approach by Wasserman. Please have a look at Related Work, Section 3, line 237ff. Furthermore, we added further details about the phase prediction result in comparison to the state of the art request based approaches by Gutterman. These approaches are described in Related Work, Section 3, line 255ff. We have chosen the phase prediction here since it is an important metric in our work and the related work.

C07: “Authors aim to demonstrate that a lighter uplink data-based approach is competitive in comparison with already proposed heavier downlink (or full packet traces) based approaches. To test some of the state of the art proposals using the same authors’ Youtube data set will allow to appreciate the contribution of the present work.”

Answer: For the best possible comparability, we included the bandwidth scenarios of reference works to our dataset. We used 300 kbps, 1 Mbps, 3 Mbps, or 5 Mbps, and 20 Mbps like mentioned in Wassermann et. al. [14], to compare to state of the art downlink/full packet trace approaches measured with mobile devices.

We hope that these revisions have adequately addressed your comments.

Round 2

Reviewer 1 Report

The authors have considered my concerns and made improvements.

Reviewer 3 Report

in general, the most part of my comments have been correctly addressed.